# Normalizing and denoising protein expression data from droplet-based single cell profiling

Matthew P. Mulè [1,2,4], Andrew J. Martins [1,4] & John S. Tsang [1,3 ✉]

Multimodal single-cell profiling methods that measure protein expression with oligo-conjugated antibodies hold promise for comprehensive dissection of cellular heterogeneity, yet the resulting protein counts have substantial technical noise that can mask biological variations. Here we integrate experiments and computational analyses to reveal two major noise sources and develop a method called "dsb" (denoised and scaled by background) to normalize and denoise droplet-based protein expression data. We discover that protein-specific noise originates from unbound antibodies encapsulated during droplet generation; this noise can thus be accurately estimated and corrected by utilizing protein levels in empty droplets. We also find that isotype control antibodies and the background protein population average in each cell exhibit significant correlations across single cells, we thus use their shared variance to correct for cell-to-cell technical noise in each cell. We validate these findings by analyzing the performance of dsb in eight independent datasets spanning multiple technologies, including CITE-seq, ASAP-seq, and TEA-seq. Compared to existing normalization methods, our approach improves downstream analyses by better unmasking biologically meaningful cell populations. Our method is available as an open-source R package that interfaces easily with existing single cell software platforms such as Seurat, Bioconductor, and Scanpy and can be accessed at "dsb [https://cran.r-project.org/package=dsb]".

[1] Multiscale Systems Biology Section, Laboratory of Immune System Biology, National Institute of Allergy and Infectious Diseases (NIAID), National Institutes of Health (NIH), Bethesda, MD, USA. [2] NIH-Oxford-Cambridge Scholars Program, Department of Medicine, University of Cambridge, Cambridge, UK. [3] NIH Center for Human Immunology (CHI), National Institutes of Health (NIH), Bethesda, MD, USA. [4] These authors contributed equally: Matthew P. Mulè, Andrew J. Martins. ✉email: john.tsang@nih.gov

Recent developments in multimodal single cell analysis involve using DNA barcoded antibodies to simultaneously profile surface proteins together with the transcriptome (e.g., CITE-seq) and/or chromatin accessibility (e.g., ASAP-seq) in single cells[1–4]. This greatly enhances our ability to discover, define, and interpret cell types and states, particularly those comprising the immune system given extensive existing knowledge connecting surface protein profiles to immune cell subsets and functions[5]. Droplet-based sequencing of single cells stained with DNA-barcoded antibodies provides a readout of protein levels in the form of antibody-derived tag (ADT) counts for each protein. This "cytometry via sequencing" approach bypasses spectral interference inherent in fluorescence-based cytometry methods, thus enabling simultaneous profiling of hundreds of proteins in single cells. While low-level normalization and modeling approaches for single cell RNA-seq data have received considerable attention[6–12], those for protein/ADT are in their infancy and more importantly, the extent and sources of noise have not been quantitatively analyzed despite the substantial levels of apparent noise reported in raw protein counts[2].

Stochastic processes during single cell mRNA capture and sequencing contribute to sampling noise[13,14] and other technical variations leading to reduced UMI counts, including zero counts for genes despite actual mRNA expression in a given cell. Such noise can be modeled with statistical distributions[15–17] or normalized, for example, by standardizing the total number of mRNA reads between cells commonly performed via scaling factors computed from each cell's total mRNA "library size"[18] (defined herein as the total UMI count for a given assay/data modality in each cell). However, these methods are not appropriate for surface protein count data for several reasons. First, a major noise component of ADT data appears to be added background noise because cells tend to have positive counts for multiple classes of proteins that are reported to be mutually exclusively expressed in distinct cell subsets. For example, compared to sparse mRNA counts, only two 0 values are present across more than 11,000 cord blood cells stained with 13 surface proteins in the original report of the CITE-seq method[2]. Second, current methods/experiments still measure only a small fraction of unique proteins with a wide range of antigen density on different cell types, resulting in individual protein counts in single cells spanning ~2–3 orders of magnitude (e.g., <10 to >1000); differences in total protein counts between cells therefore depend on the specific antibody panel used. Finally, the total protein counts detected on a given cell may reflect both technical but also biological variations such as cell size across cells and cell types, especially given the dependence of the total ADT counts on the specific antibody panel used.

The original developers of CITE-seq normalized ADT data by using a centered log ratio transformation (CLR). The resulting values can be interpreted as either a natural log ratio of the count for a given protein relative to the other proteins in the cell (CLR "across proteins", as implemented in the original report of CITE-seq[2]) or relative to other cells (CLR "across cells", a modification used in later work by the authors[19], which renders CLR less dependent on the composition of the antibody panel). The CLR transformation helps to better separate cell populations, but it does not directly estimate and correct for specific sources of technical noise including the apparent background noise mentioned earlier. The authors accounted for protein-specific noise in human cells by spiking in mouse cells to set a per-protein cutoff for determining whether a CLR transformed (across-protein) expression value was above that in mouse cells[2]. This approach appears not adopted beyond its original use, likely because it entails more complex experiments and analyses. More recent reports applied other approaches, for example, fitting models to estimate background and foreground distributions for each protein without using spike-in control cells[20,21], or using isotype antibody controls to estimate background[3,22]. It is unclear the extent to which these approaches remove noise versus biologically relevant signals since the noise sources remain unidentified; some proteins also have multimodal distributions across cells, while isotype controls are not typically used in flow cytometry for quantitative thresholding[23] since their level can reflect both biological and technical variations. Thus, determining the major sources of noise and developing dedicated methods to account for them are major unmet needs given the swift adoption and proliferation of multimodal single cell profiling methods involving the measurement of protein expression with DNA barcoded antibodies.

Here we perform experiments and computational analyses to reveal two major components of protein expression noise in droplet-based single cell experiments: (1) protein-specific noise originating from ambient, unbound antibody encapsulated in droplets that can be accurately estimated via the level of "ambient" ADT counts in empty droplets, and (2) droplet/cell-specific noise revealed via the shared variance component associated with isotype antibody controls and background protein counts in each cell. We develop an R software package, "dsb" (denoised and scaled by background), the first dedicated low-level normalization method developed for protein ADT data, to correct for both of these noise sources without experimental modifications. Our application of this approach to our own and several external data sets spanning multiple technologies and assay types demonstrates the generalizability of dsb to enhance downstream analysis, including manual and unsupervised protein-based and multimodal (joint protein–mRNA) identification of cell populations and states.

## Results

**Analysis of unstained cells reveals ambient antibody capture as a major source of protein-specific noise.** To assess protein count noise, we first utilized our previously reported dataset measuring more than 50,000 peripheral mononuclear cells (PBMCs) from 20 healthy human donors[24] stained with an 87 CITE-seq antibody panel (including four isotype controls; Totalseq-A reagents, Biolegend). Consistent with the original CITE-seq report[2], we noticed non-zero counts for most proteins in each cell, resulting in positive counts even of markers not expected to be expressed in certain cell types. We also noticed non-zero, "ambient" protein counts in tens of thousands of empty droplets containing capture beads without cells, which emerge naturally due to Poisson distributed cell loading, reminiscent of cell-free RNA observed in droplet-based single cell RNAseq[25–27]. We reasoned that background noise in CITE-seq data may partly reflect such unbound, ambient antibodies captured in droplets. To assess whether counts in empty droplets indeed reflect the ambient component in cell-containing droplets, we compared background protein levels in cell-free droplets with droplets capturing unstained control cells spiked into the cell mixture after cell staining and washing but prior to droplet generation (Fig. 1a). We found positive protein counts even for unstained control cells, and that the average log-transformed level per protein in empty droplets and unstained control cells were highly correlated (Fig. 1b). A similarly strong correlation was observed between the average protein counts in subpopulations of stained cells "negative" for a given protein and those in empty droplets (Supplementary Fig. 1a–c; negative cells correspond to those in the fraction with lower expression of the protein—see the "Methods" section), further suggesting that the noise component correlated across cells is dominated by ambient antibody capture. Thus, protein

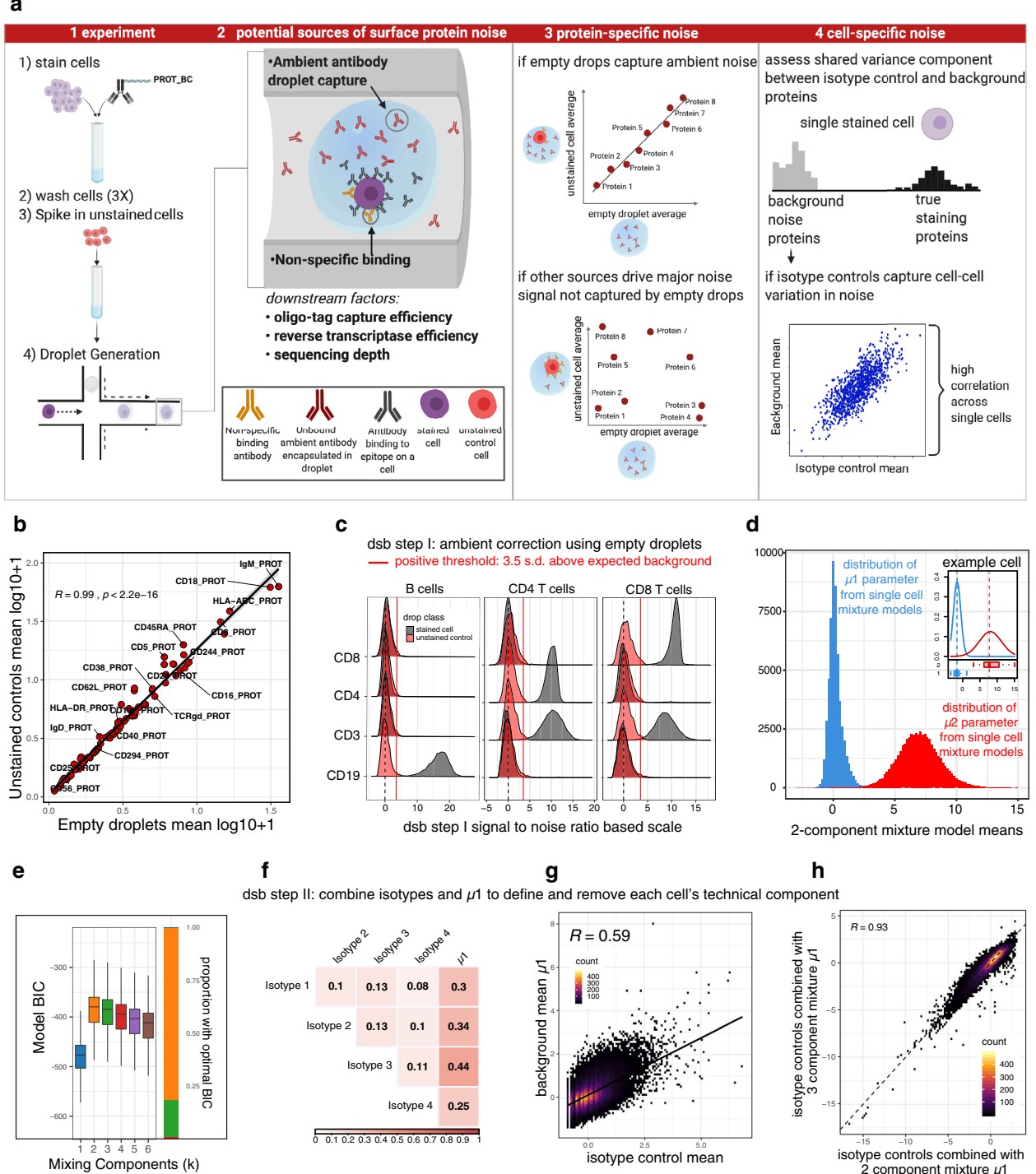

counts in empty droplets, which are available in all single-cell droplet experiments, provide a direct estimate of the ambient background due to free antibody capture for each protein. Consistent with our findings on ambient antibody capture as the major source of background noise in CITE-seq data, a recent study reporting CITE-seq antibody titration experiments across a` wide concentration range demonstrated that background noise increased with the antibody staining concentration, with some antibodies at or above 2.5 μg/mL having even more cumulative UMIs in the empty droplets compared to cells[28]. Our observation thus motivated the first step of our method to remove protein-specific technical noise: transforming counts of each protein in cell-containing droplets by subtracting the mean and

dividing by the standard deviation of that same protein across empty droplets (see the "Methods" section). The resulting transformed protein expression values for each cell reflect the number of standard deviations above the expected ambient capture noise, thus centering the negative cell population for each protein around zero to help improve interpretability of the resulting protein expression values (Fig. 1c).

**Shared variance between isotype controls and background protein counts in single cells provide cell-intrinsic normalization factors.** In addition to ambient noise correlated across single cells as captured by average readouts from empty droplets,

**Fig. 1 Antibody-derived protein UMI count data noise source assessment. a** 1 and 2: Experimental setup and potential noise sources in CITE-seq data. 3: protein-specific noise: if ambient antibody encapsulated in droplets constitutes a major source of protein-specific noise, values should be highly correlated with those in unstained control cells (top); if control cells contain information on noise not captured by empty drops, the correlation should be weak. 4: Cell-specific noise evaluated through the correlation between the background protein population mean and isotype controls across single cells. Created with BioRender.com. **b** Average protein log10(count + 1) of unstained control cells spiked into the stained cell pool prior to droplet generation (y-axis) versus that of droplets without a cell (x-axis). Pearson correlation coefficient and p value (two sided) are shown. **c** Density histograms of protein expression of lineage-defining proteins within major subsets in stained cells (black) and unstained controls (red) normalized together using dsb step I (ambient correction and rescaling based on levels in empty droplets). **d** A two-component Gaussian mixture model was fitted to the protein counts within each single cell; the distributions of the component means from all single cell fits (blue = "negative" population; red = "positive" population) are shown, protein distributions from a randomly selected cell shown in the inset. **e** Comparison of Gaussian mixture models fit with between $k = 1$ and $k = 6$ subpopulations to dsb normalized protein values for $n = 28{,}229$ cells from batch 1 after dsb step I (ambient correction) but prior to step II, vs. the model fit Bayesian Information Criteria (BIC, using mclust R package definition of BIC where larger values correspond to a better fit) from the resulting 169,374 models. Boxplots show the median with hinges at the 25th and 75th percentile, whiskers extend plus or −1.5 times the inter quartile range. $k = 2$ component Gaussian mixtures have the best fit in more than 80% of cells (orange, right inset bar plot). **f** Pearson correlation coefficients among isotype controls and background component mean inferred by Gaussian mixture model ($\mu 1$ fitted per cell as in **d**); all corresponding p values (two sided) are <2e−16. **g** Scatter density plot between $\mu 1$, the mean of each cell's negative subpopulation from the per-cell Gaussian mixture model (blue in **c**) versus the mean of the four isotype controls across single cells. Pearson correlation coefficient is shown (two-sided p value < 2e−16). **h** The distribution of the dsb technical component as calculated using a 2 component (x-axis) vs. 3 component (y-axis) mixture model to define the $\mu 1$ parameter, Pearson correlation coefficient, p value (two-sided) < 2e−16.

cell/droplet-intrinsic technical factors including but not limited to oligo tag capture, cell lysis, reverse transcriptase efficiency, sequencing depth and non-specific antibody binding, can contribute to cell-to-cell variations in protein counts that should ideally be normalized across single cells. Given that the differences in total protein UMI counts between individual cells could reflect biologically relevant variations, such as those due to the physical size of naïve vs. activated lymphocytes, library size normalization (dividing each cell by the total library size) could remove biological rather than technical cell to cell variations. In addition, since current CITE-seq antibody panels are a small subset of total surface proteins, the assumption that total UMI counts should be similar among cells may not be valid. Here we integrated two types of independently derived measures to reveal a more conservative (i.e., avoiding over-correction and removal of biological information), robust estimate of the factor associated with cell-intrinsic technical noise (Fig. 1a).

First, the four isotype control antibodies with non-human antigen specificities in our panel could in principle help capture contributions from non-specific binding and other technical factors discussed above. The counts of the isotype controls were only weakly (but significantly) correlated with each other across cells (Fig. 1f), and interestingly, the correlation between the mean of four isotype controls and the protein library size (which has both biological and technical components) across single cells was even higher (Pearson correlation 0.45) than that between the protein library size and the individual isotype controls (average Pearson correlation 0.25). This suggests that while each isotype control may be individually noisy, and their levels may still partially reflect biological contributions, collectively their shared component of variation (i.e., as reflected by their average) may better capture technical noise in the experiment. Second, to further boost the robustness of estimating cell-intrinsic technical noise, particularly given that the number of isotype controls available in practice can be limited, we sought an additional estimate of droplet-intrinsic technical variation. Since each cell in a sample of multiple distinct cell types (e.g., PBMCs) is expected to express only a subset of protein markers in staining panels, we reasoned that the distribution of each cell's non-staining proteins (e.g., those specifically expressed in other cell types/lineages) could be differentiated from the cell's "positively expressed" proteins by fitting a 2-component mixture model to each cell. If so, the average counts in the population of non-staining/negative proteins could reflect and therefore serve as another readout of

the cell's technical component that could then be integrated with the cell-intrinsic noise captured by isotype controls. To assess this hypothesis, we applied a Gaussian mixture model with two ($k = 2$) subpopulations to fit the protein counts within each single cell after correcting for the protein-specific ambient noise we identified above (see below and the "Methods" section; Fig. 1d). We found clear separation between the background (with mean = $\mu 1$) and positive (mean = $\mu 2$) protein population with substantial cell-to-cell heterogeneity of subpopulation means (Fig. 1d). We next assessed the robustness of using a two-component mixture to model the protein counts of individual cells by comparing $k = 1$–6 component models assessed using the Bayesian Information Criterion (BIC). While two-component models had the best fit in a majority (81%) of cells, indicating a bimodal protein distribution within single cells, $k = 3$ models had the best fit in nearly all remaining cells (Fig. 1e, Supplementary Figs. 2a, b; see also Supplementary Note). The BIC for these cells were very similar to the corresponding $k = 2$ models (Supplementary Fig. 2c), indicating that the two-component fits were identifying very similar positive and negative populations. Importantly, for the minority of cells with optimal $k = 3$ or 4 models, the resulting mean of the lowest expression population ($\mu 1$ estimate) was highly concordant when the same cells were fit with a $k = 2$ model (Supplementary Fig. 2d–f). These data suggest that a two-component Gaussian mixture fit of the protein population within single cells can robustly delineate the negative background protein count population for most cells.

Together $\mu 1$ and the isotype controls provide estimates of technical noise within each single cell. However, each variable may be individually noisy; we thus assessed information sharing among these variables. The correlations between $\mu 1$ and each individual isotype control (average correlation $r = 0.33$) or the average of all four isotype controls ($r = 0.59$) were higher than those between the isotype control themselves (average correlation $r = 0.11$), suggesting that the shared variation (i.e., average) between the independently inferred $\mu 1$ and isotype controls captured unobserved, latent factors contributing to technical noise (Fig. 1f, g). We thus reasoned that the first principal component score ($\lambda$) capturing the shared variation of $\mu 1$ and the isotype controls across single cells would be a robust measure of technical noise intrinsic to individual cells. $\lambda$ was associated with the protein library size across single cells within cell clusters (Supplementary Fig. 3a clusters defined after dsb steps I and II, see the "Methods" section), supporting the notion that $\lambda$ captures

the technical component of the protein library size. Furthermore, consistent with the observation above regarding the Gaussian mixture model fit, $\lambda$ was highly concordant regardless of whether the background ($\mu 1$ estimate) was defined using a $k = 2$ or $k = 3$-component Gaussian mixture (Fig. 1h).

Given the information sharing between $\mu 1$ and isotype controls, we recommend the inclusion of multiple isotype controls in CITE-seq experiments to serve as anchors for robust inference of technical normalization factors (see Supplementary Note). Together, our data indicate that while the signal from individual measures, such as isotype controls can be noisy and may reflect multiple yet often unknown sources of variation, their correlated component of variation can serve as a robust normalization factor for surface protein expression in single cells. Thus, in a second, optional but recommended step, our method computes $\lambda$ for each cell as its "technical component", which is then regressed out of the ambient noise corrected protein values (Fig. 1c) generated by step 1 above (see the "Methods" section). The underlying modeling assumptions of the dsb technical component also held well in seven independent datasets generated via different assay platforms and protein panels of diverse sizes (from 17 to more than 200 proteins; see below).

**Comparison with other transformations and assessing dsb in independent datasets generated by different technology platforms.** The unstained spike-in cells above should reflect the level of protein specific, "ground-truth" noise, we thus used these cells to visually compare dsb with other normalization transformations (Supplementary Fig 3e; see the "Methods" section). Unstained cells normalized using dsb centered around zero, while CLR or log transformation placed these cells at arbitrary locations. For example, CD4 has a trimodal distribution due to absence of expression in populations such as B lymphocytes, low expression in CD14+ monocytes and high expression in helper T cells; dsb normalized values centered the background population together with unstained control cells at zero and delineated low-level CD4 staining on monocytes. In contrast, these monocytes are closer to and partially overlapped with the unstained population when CLR or log normalization were used (Supplementary Fig 3e). We further compared dsb to CLR (the version that normalizes across cells) since CLR is the most commonly applied transformation for ADT data to date and normalization across cells should depend less on the protein staining panel than CLR across proteins. Using $k$-medoids clustering of single cells based on protein expression data only, the Gap-Statistic[29], which reflects improvement in within-cluster coherence relative to that expected of random data drawn from a reference distribution, was consistently higher using dsb compared to CLR across different values of $k$. However, the trend as a function of $k$ was similar between dsb and CLR, suggesting that the improvement could be partly due to scaling differences between these two transformations (Supplementary Fig. 3f). Finally, differential expression analysis comparing major immune cell populations with the rest of the cells revealed that key lineage and cell-type-specific proteins (e.g., CD56 on NK cells) tended to have larger fold changes when using dsb normalized protein values compared to CLR (Supplementary Fig. 3g).

We next tested the general applicability of dsb by using several independent, publicly available CITE-seq datasets. We first assessed whether the modeling assumptions developed using our own CITE-seq data would generalize to four other CITE-seq datasets that profiled ~5000 to 10,000 cells using 14–29 surface phenotyping proteins and three isotype controls, and were generated using different versions of the 10X Genomics droplet profiling kit than the one we used. Similar to our dataset, we

detected a large number of empty droplets containing antibody reads (>50,000) inferred by the EmptyDrops[25] algorithm used in the Cell Ranger barcode rank algorithm; the number of cell-containing droplets estimated by Cell Ranger and further filtered by quality control metrics (3000–8000 droplets) was also consistent with the number of loaded cells (Fig. 2a, Supplementary Fig. 4a, h, o). Thus, protein-specific ambient noise can be estimated as in our data set using these empty droplets. Applying dsb without any modification resulted in biologically interpretable protein-based clusters (Fig. 2b, c, Supplementary Fig. 4e, f, l, m, s, t) and canonical immune cell populations could be clearly delineated by conventional biaxial plots (Fig. 2d, Supplementary Fig. 4g, n, u). Importantly, the model-fitting behavior and correlations among isotype controls and background counts observed in our dataset were similarly observed in these independent datasets, including: (1) The $k = 2$ component Gaussian mixture model had the best fit according to BIC in most single cells (Fig. 2e, 89% average across four CITE-seq datasets); (2) the estimated $\mu 1$ (mean of background protein counts) for each cell correlated significantly with the mean of isotype controls across single cells and was higher than the correlation with individual isotype controls (Fig. 2f, g, Supplementary Fig. 4b, c, i, j, p, q); (3) the inferred technical component using isotype controls and $\mu 1$ was correlated with the protein library size (Fig. 2h, Supplementary Figs. 4d, k, r); finally, (4) even on the smallest panel (14 phenotyping antibodies, 3 isotype controls) the per cell technical component $\lambda$ was highly concordant regardless of whether the background ($\mu 1$ estimate) was defined using a $k = 2$ or $k = 3$-component Gaussian mixture (Supplementary Fig. 2g).

We next tested the applicability of dsb to several new types of multimodal single cell data generated by technologies that measure surface protein expression in droplet captured single cells using oligo-barcoded antibodies including (1) "proteogenomic" data (protein + DNA mutation assays from Mission Bio; 9 proteins plus an isotype control), (2) ATAC-seq with Select Antigen Profiling (ASAP-seq: protein and chromatin accessibility; 238 proteins plus isotype controls), and 3) Transcription, Epitopes, and Accessibility (TEA-seq: protein + chromatin accessibility and transcriptome assessment; 45 proteins plus one isotype control). All datasets had ADT reads in a large number of empty droplets (Supplementary Figs. 5a, d, e). Our method was compatible with the proteogenomic dataset, helping to identify markers for each cell cluster after correcting for protein-specific background levels estimated from >16,000 empty droplets (Supplementary Figs. 5a–c). In the ASAP-seq dataset that measured multiple isotype controls, $\mu 1$ again correlated significantly with the mean of isotype controls across single cells and this correlation was higher than that among the individual isotype controls (Supplementary Figs. 5f, g), and the inferred per-cell dsb technical component was correlated with the library size as observed above (Supplementary Fig. 5h). In TEA-seq and ASAP-seq data, the negative staining cells could often be identified by applying the same 3.5 threshold that we applied in our and other data sets (Supplementary Figs. 5i–k and see below). The compatibility and utility of dsb with large protein panels such as in the ASAP-seq dataset is consistent with our recent CITE-seq analysis of Covid-19 patients using a similarly large panel where dsb helped enable accurate cell population identification by both automated clustering and manual gating[30]. A summary of results from these datasets is shown in Fig. 2i.

**Case study I: dsb improves interpretation of protein-based and joint protein–mRNA clustering results.** We next further investigated the ways in which normalization with dsb could help improve cell type identification. By design, dsb zero-centers the background population for each protein and provides normalized

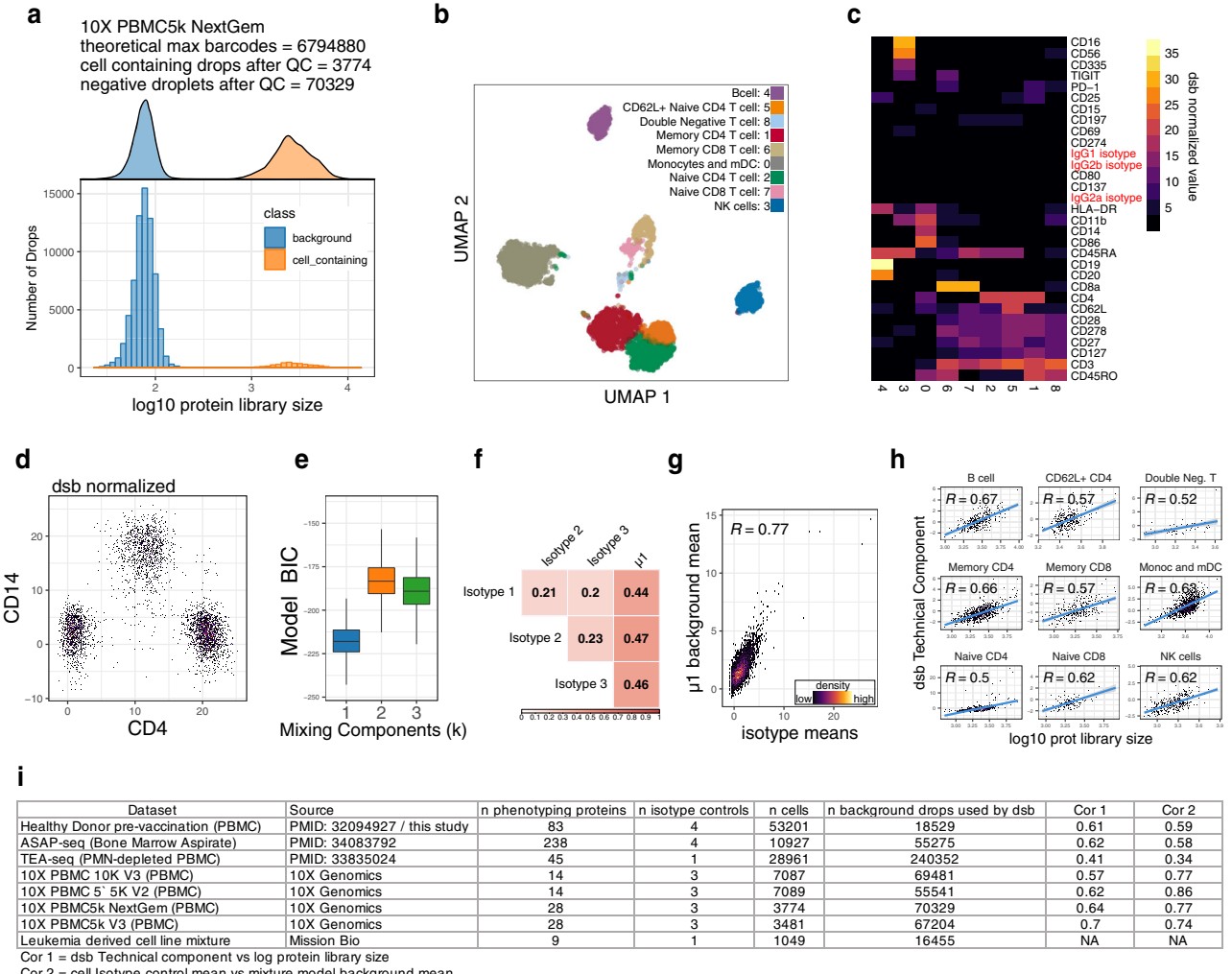

**Fig. 2 Assessment of dsb model assumptions and performance of dsb normalization on external datasets.** Panels **a**–**h** Application of dsb to a publicly available dataset generated using 10X genomics "NextGem" chemistry measuring 29 proteins across ~5K cells. **a** The protein library size distribution of empty and cell-containing droplets used for dsb normalization. **b** UMAP of single cells based on dsb normalized protein values with colors representing clusters obtained from clustering cells on dsb normalized protein values. **c** Heatmap of the average of dsb normalized values per protein-based cluster shown in (**b**). **d** The distribution of CD14 and CD4 dsb normalized values. **e** As in Fig. 1e, Gaussian mixture model parameters fit to the dsb normalized values of each single cell after step I (ambient noise/background droplet based correction). The Bayesian Information Criterion (BIC) of the model vs. number of components in the model fit for each cell ($n = 3774$ cells). Boxplots show the median with hinges at the 25th and 75th percentile and whiskers extending plus or minus 1.5 times the inter quartile range. **f** As in Fig. 1f, Pearson correlation coefficient matrix of variables used to define each cell's technical component; each isotype control and $\mu1$, the Gaussian mixture model background mean across proteins for each cell. **g** As in Fig. 1g, Pearson correlation coefficient between the inferred cell-specific background mean $\mu1$ from the Gaussian mixture model vs. the mean of isotype controls in each cell. **h** The relationship between each cell's technical component and the cell's protein library size (Pearson correlation coefficient shown as in Supplementary Fig 3a with 95% confidence interval in gray). **i** Summary statistics for the eight independent datasets assessed in this study; Cor 1 and 2 correspond to the Pearson correlation coefficient for assessing the relationships between variables shown in (**h**) and (**g**) across cells for each dataset.

expression interpretable as signal above expected background noise. These features are thus particularly helpful in manual gating across cell lineages (Supplementary Fig. 6a) and can improve the annotation of cell types derived from unbiased clustering. In contrast, distinguishing true biological expression from noise within individual cell clusters can be challenging when using transformations such as the CLR, partly because CLR protein values lie on a non-zero-centered scale (each protein also has a distinct noise floor); therefore, cells can appear to express markers known to be specific for other cell lineages. For example, in cluster 4 from our PBMC data (framed cluster in Fig. 3a), proteins such as IgA/IgM and CD57 could be mis-interpreted as showing signal above noise (Fig. 3b). In contrast, dsb normalized values for IgA, IgM, and CD57 are zero-centered (Fig. 3b),

indicating that the level of these proteins in this cluster was statistically similar to the level in empty droplets and were therefore not expressed (Fig. 3c, d—red proteins). In contrast, CD16, CD244, and CD56 had dsb values above 8 (i.e., >8 standard deviations above the mean in empty droplets, +/− the correction from regressing out the technical component), suggesting these were CD57 negative CD16++ CD56+ NK cells, which are not known to express B-cell markers such as IgM or IgA. In general, cell clusters identified using dsb normalized protein values had cell type-defining proteins detected above the same threshold (3.5) applied within each cell cluster (Fig. 3e, Supplementary Fig. 6b, c).

We also assessed compatibility of dsb with an unsupervised joint mRNA-protein clustering algorithm that constructs a

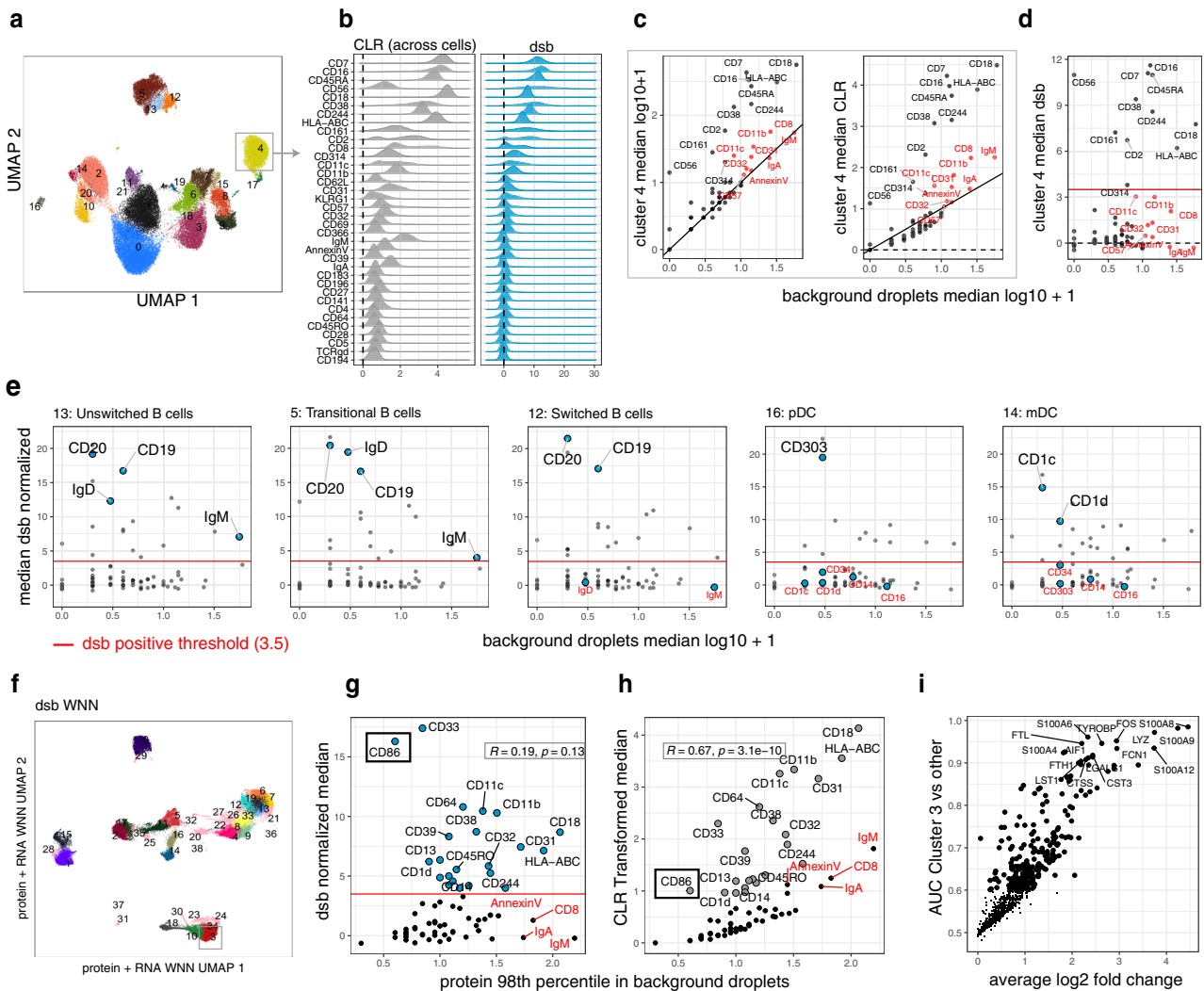

**Fig. 3 Case study I: dsb improves interpretation of cell clusters derived from protein-based and joint mRNA–protein clustering. a** UMAP plot of single cells labeled by cluster number (clustering was performed using dsb normalized protein values). **b** The distribution of protein expression of cluster 4 (highlighted with a gray box in (**a**)) using CLR (across cells) or dsb for normalization. **c** Median log +1 protein levels (left) and CLR transformed across cells (as in (**b**), right) in cells from cluster 4 versus the level in empty droplets; proteins highlighted in red are comparable in expression to "positive" proteins after log transformation (left) and CLR transformation across cells (right) but are similar to background levels in empty droplets (identity line $y = x$ shown in black). All proteins with median log10 expression >1 but <3.5 after dsb normalization are labeled with the protein name. **d** Similar to (c), but the y-axis shows the median dsb normalized values; proteins in red (those near the diagonal in (**c**)) are now residing below our uniformly applied dsb positivity threshold of 3.5, reflective their proximity with mean counts in empty droplets; proteins above the red line have median dsb normalized expression within the highlighted cluster 4 (see (**a**) and (**b**)) above 3.5, i.e., 3.5 standard deviations above ambient noise, ±adjustment for the cell intrinsic technical component. **e** The dsb normalized value vs. the median value in empty droplets of proteins within a subset of protein-defined clusters. A subset of proteins informative for cluster identification from B cell and dendritic cell subsets with a dsb value above 3.5 (red line) are annotated with the protein name within each panel and are labeled in red when below 3.5 within each subset. Proteins labeled for B cell subsets (C13: Unswitched B cells, C5 Transitional B cells, C12 Switched B cells) include B cell proteins CD20, CD19, IgD, and IgM, proteins labeled for the dendritic cell subsets (C16: pDC, C14: mDC) include innate cell markers CD1c, CD1d, CD34, CD14, CD16, and CD303. **f** UMAP plot of the same cells shown in (**a**) but the UMAP embeddings and clusters here were derived using Seurat's weighted nearest neighbor (WNN) mRNA-protein multimodal algorithm applied to dsb normalized values. **g** Similar to (**d**) but derived using cells from WNN cluster 3; Pearson correlation coefficient and p value (two sided) are shown between median dsb normalized values and the 98th percentile expression value (log10) of the same protein in empty droplets. **h** Similar to (**g**) but for CLR normalized values. **i**. Differentially expressed genes (ROC test; see the "Methods" section) for cell in cluster 3 vs. other clusters.

weighted nearest-neighbor (WNN) joint embedding of CITE-seq mRNA and protein data[31] (Fig. 3f). We ran WNN clustering using the same processed mRNA data together with ADT data normalized by either dsb or CLR (across cells). The clustering results were similar, suggesting dsb and CLR led to broadly concordant results. However, closer examination of individual clusters revealed that dsb could lead to more interpretable results. Notably, CD14-positive cells (presumably monocytes) were

distributed across multiple dsb-derived clusters, including cluster 3 characterized by elevated CD86 (Fig. 3g). In contrast, the CLR value of these same cells was relatively low for CD86 but high for other markers (e.g., CD8 and IgM) that should not be expressed by monocytes (Fig. 3h). Furthermore, median CLR values in these cells (but not dsb—Fig. 3g) were correlated with the 98th percentile of expression in empty droplets across proteins ($R = 0.67$, $p = 3.1e-10$; Fig. 3h), suggesting that protein-specific

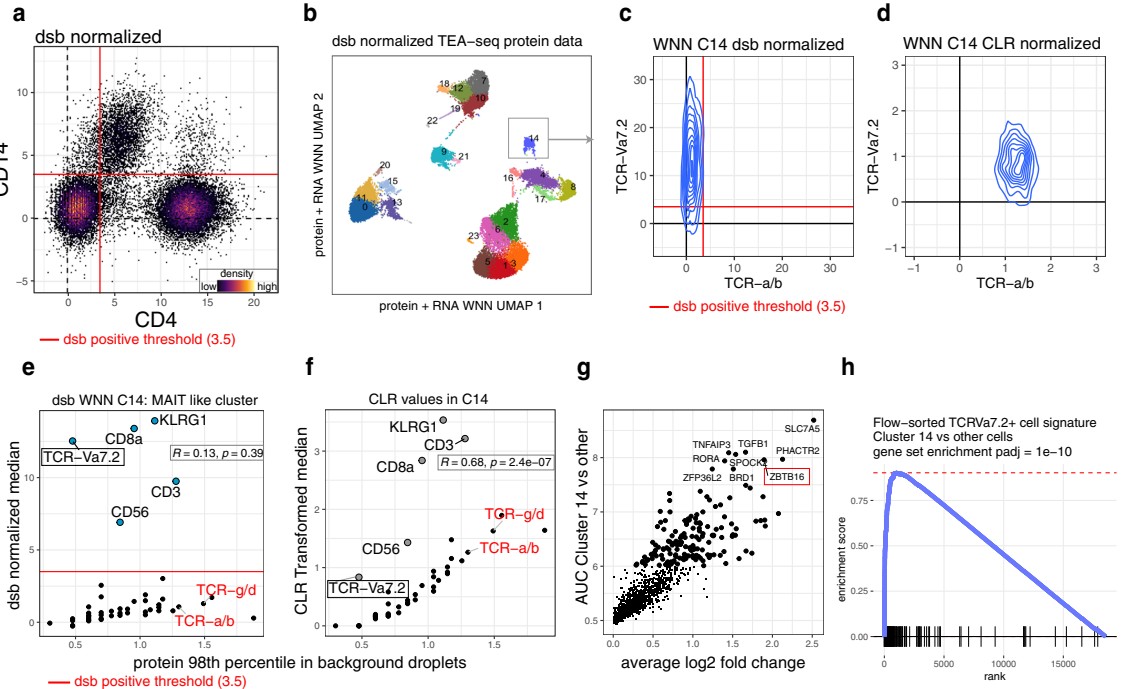

**Fig. 4 Case study II: application of dsb to tri-modal TEA-seq data unmasks a MAIT cell population obscured by noise in CLR normalization.** Analysis of TEA-seq (transcriptome, epitopes, and accessibility) tri-modal single cell assay data. **a** dsb normalization of protein data from TEA-seq showing the distribution of CD4 and CD14 with the same 3.5 threshold used throughout the study. **b** UMAP plot of single cells and clusters derived by WNN joint mRNA–protein clustering with protein data normalized using dsb. **c** Bi-axial distribution of the alpha beta and va7.2 T cell receptor (TCR) proteins in cluster 14 cells normalized by dsb and **d** the same cells' CLR normalized values. **e** Similar to Fig. 3g but here for cluster 14 from (**b**) using dsb or **f** CLR normalized values (y-axis); in both plots Pearson correlation coefficients and p values (two sided) are shown between normalized values (y axis) and values in empty droplets (x axis). **g** Differential expression analysis (ROC test) of genes in cluster 14 vs. other clusters. **h** Gene set enrichment of a MAIT cell signature constructed from FACS-sorted TCR-va7.2+/MAIT cells compared to other T cells (RNA-seq data from Park et al. 2019) with genes ranked by log2 fold change in cluster 14 cells vs. other cells as in (**g**).

ambient noise contributed substantially to the CLR values; this noise source was successfully accounted for by dsb via the use of empty droplets. Finally, relative to the rest of the cells, differentially expressed transcripts in dsb-derived cluster 3 include inflammatory and activation genes (Fig. 3i), consistent with the CD86-high phenotype revealed by dsb.

**Case study II: dsb unmasks MAIT cell population in tri-modal TEA-seq data.** As a second example, we further analyzed tri-modal transcriptome, protein, and chromatin accessibility (TEA-seq) data[32]. Visual inspection suggested improvement in biaxial plots after dsb normalization as the same interpretable threshold of 3.5 applied to all datasets in this study delineated two cell populations based on CD4 and CD14 (Fig. 4a) compared to normalization with protein library size (as implemented in the original TEA-seq study), and CLR (Supplementary Fig. 9a, b). To assess unsupervised multimodal clustering, we carried out the same comparison of CLR and dsb normalization using WNN clustering (combining mRNA and protein) as above but on TEA-seq data. Similar to above, the clustering results overlapped significantly (Chi-squared test, $p < 2e-16$ Supplementary Fig. 9c, d). However, we noticed phenotypic marker differences within a specific T cell cluster that could substantially change the biological interpretation of the resulting cell population. During thymic development, human T cells rearrange variable, diversity and joining (VDJ) genes at the T cell receptor (TCR) locus. The resulting TCR gene rearrangements are distinct to functional categories of T cells with known specialized functions. This TEA-seq data included antibodies specific for alpha-beta

(TCR a/b—conventional helper and cytotoxic T cells), gamma-delta (TCR g/d gamma-delta T cells), and Va7.2 (specific for mucosal associated invariant T (MAIT) cells). The MAIT TCR Va7.2 median dsb values were high (~15 standard deviations above background noise) in cell cluster 14 (with more than 700 cells); as expected, cells in this cluster expressed TCR Va7.2 exclusively with no other TCR proteins according to dsb normalization (Fig. 4c, Supplementary Fig. 9f). In contrast, the CLR normalized values of the cells in this cluster had higher median values for TCR a/b than TCR-va7.2; both TCRs were similarly distributed and it was thus unclear which was truly expressed given the uncertain noise floor of CLR normalized counts (Fig. 4d, Supplementary Fig. 9e). This was also the case for the gamma-delta T cell receptor protein, which was around zero after dsb normalization (Supplementary Fig. 9e, f). CD56, CD3, CD8, and KLRG1 in Cluster 14 were also positive based on dsb (more than 6 standard deviations above background noise) (Fig. 4e), thus broadly consistent with the known phenotype of CD8 + MAIT cells[33]. These cells have distinct biological functions from conventional T cells, partly due to their semi-invariant T cell receptor (TCR-Va7.2) specific for bacterial metabolic products presented via major histocompatibility complex-related protein MR1[34]. Based on CLR normalized protein levels alone, cells in cluster 14 had a phenotype resembling conventional T cells with elevated cytotoxic capacity (TCR a/b, KLRG1 and CD56 positive)[35,36]. Since dsb corrects for protein-specific noise, we hypothesized that the apparent expression of both TCRs in cluster 14 after CLR normalization was likely due to ambient noise present in CLR transformed data. Supporting this notion, the median CLR values (but not the dsb-derived values) were

correlated with the 98th percentile values from empty droplets (Pearson correlation 0.8, two-sided $p = 2.4e-7$, compared Pearson correlation 0.13, two-sided $p = 0.39$ for dsb), and both the alpha-beta and gamma-delta TCR proteins were among the highest ranked proteins based on expression in empty droplets (Fig. 4e, f). To further assess the identity of this cluster, we performed unbiased differential mRNA expression analysis of cluster 14 cells versus other clusters (Fig. 4g). Among the top discriminative markers for cluster 14 was the transcription factor ZBTB16 (Fig. 4g), which is known to be elevated during iNKT and MAIT cell differentiation[37], expressed by mature MAIT cells[38,39], but suppressed during conventional naïve T cell differentiation[40]. We next constructed a 165-transcript MAIT cell signature derived from the top differentially expressed genes reported in an independent study, which used bulk RNA-seq to compare FACS-sorted TCR-Va7.2+ human MAIT cells versus other T cells lacking this TCR[41]. This MAIT cell signature was significantly enriched (Fig. 4h) in differentially expressed genes from cluster 14 (normalized GSEA enrichment score 2.64, $p$ value $1e-10$). As this example demonstrates, dsb helped to avoid potential misannotation of a T cell subset and revealed biologically coherent mRNA and protein profiles of MAIT cells. Thus, dsb is compatible with and can improve downstream analysis outcomes of multimodal single cell data such as TEA-seq.

## Discussion

Our experiments and computational analyses revealed ambient capture of antibodies by droplets is a major source of protein-specific noise in droplet-based ADT data. Our method, dsb, estimates and corrects this noise component without experimental modifications since we found that it can be reliably estimated using empty droplets, which are abundant in droplet based single cell datasets. On top of protein-specific noise, cell intrinsic noise was apparent given our observation of the strong correlation (i.e., shared variance) among distinct isotype controls and the average ADT level of background proteins inferred by mixture modeling within single cells. This correlated component affords dsb to implement a conservative approach to estimate and correct for cell-to-cell technical noise, an improvement over approaches that use individual isotype controls or total protein library size because individual variables alone are inherently noisier and could contain more biological (as opposed to technical) signals. We found that application of dsb to both our own and independent multimodal single cell datasets with ADT data improved the identification and annotation of cell types and states based on protein-based or multimodal clustering approaches.

Recent methods proposed to use joint probabilistic modeling of mRNA and protein[21,42] with one of the goals being identification of protein expression above noise. For example, TotalVI[42] uses an mRNA and protein generative neural network model to estimate posterior probability distributions of protein expression, which identified cells with zero, low, or high probability of CD4 protein levels in human PBMCs. As expected, this identified monocytes and T-helper cells based on known low and high surface CD4 protein levels on these cells, respectively; these populations were similarly recovered by dsb normalized populations. While such end-to-end probabilistic models hold promise for single cell analysis, the TotalVI counts are denoised in non-normalized UMI count space—to use these raw UMI counts for downstream analysis tasks outside the probabilistic neural network framework, the values would still need to be normalized, for example via a log transformation. Such probabilistic models are thus complementary to and distinct from dsb, which focuses on low-level protein- and cell-intrinsic denoising and normalization unique to ADT protein data by directly inferring and removing the two

noise components detailed in our analyses above. In addition, the specific noise sources revealed by our analyses and approaches to estimate them could lead to more informative prior distributions used by Bayesian probabilistic modeling approaches such as TotalVI. As demonstrated here, the denoised and normalized data from dsb can be used in any downstream analysis application to potentially enhance the results of higher level single cell data analysis methods, such as joint protein–mRNA clustering[31,43–45].

We further detail the experimental evidence for noise sources as well as the modeling assumptions, caveats, and limitations of our method in the Supplementary Note. Briefly, we assessed (1) the robustness of our estimation of protein-specific noise, (2) the sensitivity of dsb normalized values to different methods of defining empty droplets, (3) the impact of different cutoffs for defining background droplets for use with dsb, (4) normalization across batches: normalizing multiple experimental batches together vs. applying normalization separately to each batch, and (5) caveats for using dsb on datasets without isotype control antibody measurements. The use of different methods for defining background droplets had negligible impact on normalized expression values, however, defining a reasonable subset of barcodes as background droplets still requires care. The dsb package documentation provides code to extract and quality control the background droplet population from the raw data matrix. It uses all cell barcodes from the Cell Ranger alignment tool by default, although other alignment tools such as kallisto[46] and CITE-seq-Count[47] can also be used. In our own dataset used above, we also found little differences in dsb-normalized expression values between first merging data across from batches before applying dsb vs. applying dsb to each batch individually. However, in general this could be dependent on the extent of uniformity among the batches. Finally, additional analysis further supported the benefit of including isotype controls to help correct for cell-to-cell technical noise in step II of dsb (see Supplementary Note for details).

The dsb package is computationally efficient and can process on the order of $10^5$ cells on a laptop, e.g., the primary dataset in this study (with >53,000 cells) was normalized and denoised in under 4 min. The output can be easily integrated with diverse single cell software platforms such as Bioconductor[48], Seurat[49], and Scanpy[50].

## Methods

**The denoised scaled by background normalization (dsb) method**. The dsb method is implemented via the R package "dsb [https://cran.r-project.org/package=dsb]" through a single function call to *DSBNormalizeProtein()*, which models and accounts for (1) protein-specific ambient noise correlated across single cells as captured by average readouts from empty droplets and (2) droplet/cell-specific technical noise revealed via the shared variance component associated with isotype control antibodies and background protein counts in each cell. Internally the function is carried out in two major steps. In step I, protein counts in empty droplets are used to estimate the expected ambient background noise for each antibody. Each protein's counts in cell-containing droplets are thus rescaled using this expected noise measurement as:

$$Y = \frac{\log(x_i + P) - \mu_n}{\sigma_n} \tag{1}$$

where $\log x_i$ is the natural log of the count for protein $Y$ in cell $i$, $P$ is a pseudocount added to prevent taking the log of zero and to stabilize the variance of small counts, and $\mu_n$ and $\sigma_n$ are the mean and standard deviation of empty droplets for protein $Y$, respectively, computed in the same way in natural log space with pseudocount $P$ added. The value of $P$ can be empirically chosen; we use 10 by default, finding this provides good clustering performance and visualization of the CITE-seq data we have analyzed. The transformed expression estimate ($Y$) for the protein in each cell can be interpreted as the number of standard deviations above the expected ambient background noise of that protein. This expression matrix can be returned without further removing technical cell to cell variations in step II, for example if isotype controls are not available, by setting *denoise.counts = FALSE* in the R function, however we strongly recommend using isotype controls and further correcting cell to cell technical variations by fitting and removing each cell's dsb

technical component in step II below by setting *denoise.counts = TRUE* and *use.isotype.control = TRUE* (the function default).

In step II, dsb denoises cell-to-cell technical variations by defining and removing the "technical component" of each cell's protein values after ambient correction from step 1. This step fits a model to each cell to learn the background population mean, and then combines this value with the shared variation across values of isotype control proteins. In the first part of this two-part step, dsb fits a Gaussian mixture model through the expectation-maximization algorithm implemented with the mclust[51] R package to the transformed count of each cell from step 1 with $k = 2$ mixture components:

$$(fx_i) = \phi_1 N_1(x|\mu_1, \sigma_1) + \phi_2 N_2(x|\mu_2, \sigma_2) \tag{2}$$

In the model above, the log normally distributed proteins of each cell $i$ comprising the non-staining noise/background protein subpopulation for that cell are estimated by ($N_1$), and $\mu 1$ is the mean of the background protein subpopulation $N_1$ in that cell. A noise variable matrix is then constructed by combining all the fitted $\mu 1$ values with the isotype control values for all cells. dsb then calculates principal component 1 (i.e. the primary latent component "$\lambda$") of these variables in the noise matrix across cells:

$$\lambda_1 = \phi_{1,1}(\mu 1) + \phi_{1,2}(\text{Isotype1}) \dots \phi_{1,p}(\text{Isotype} \, p) \tag{3}$$

where loading vectors in equation III calculated by the R function *prcomp()* are multiplied by the noise matrix, forming each cell's PC1 score $\lambda_1$ which determines the cell's "dsb technical component". Finally, the dsb technical component for each cell is then regressed out of the ambient-noise-corrected values "$Y$" from part 1; the values returned by dsb are the residuals (plus intercept) of a linear model regressing the ambient corrected values on the technical component for each protein. Internally, to implement this step dsb uses a function from the limma[52] package *removeBatchEffect()* for robust and efficient matrix decomposition to fit and then regress out the effect of a specified covariate (in this case the technical component $\lambda_1$ from Eq. (3)) from a matrix of variables (proteins) across observations (cells).

We strongly recommend using isotype controls if using the cell to cell denoising step (i.e. if setting *denoise.counts = TRUE*) as we observe that the use of more isotype controls increases the robustness of the calculation of the technical component. See the dsb software documentation on CRAN and the Supplementary Note for additional information on usage and definition of the technical component in experiments without isotype controls.

**CITE-seq on 20 human PBMC samples.** CITE-seq data analyzed here were previously used to assess the cellular origin and circuitry of baseline immune signatures[24]; an earlier version of dsb was used therein to normalize the protein data which is identical to the default method implemented in the dsb package, with exception of the pseudocount used (1 vs. 10, see below). Experiment details can be found in our prior report[24]. Briefly, oligo-labeled antibodies for sample barcoding (cell "hashing") and surface target protein detection were obtained from Biolegend. After incubating each sample with a barcoding antibody[19], cells from each donor were pooled into one tube and stained with an optimized mixture of oligo-labeled CITE-seq antibodies against target surface proteins. Two experimental batches were performed on consecutive days, using aliquots of the same pool of antibodies for each batch. The pooled donor cells from each of two batches were each distributed evenly across six lanes (per batch) of the 10x Genomics Chromium Controller using Single Cell 3′ expression reagents (version 2). Sample barcoding (HTO) and target surface protein (ADT) libraries were prepared as in the original CITE-seq report and according to the publicly available CITE-seq protocol (version 2018-02-12, cite-seq.com). cDNA libraries were prepared using the 10x Genomics v2 kit according to manufacturer's instructions. Libraries were sequenced using the Illumina HiSeq 2500 using v4 reagents. We used CITE-seq Count[47] for HTO and ADT read mapping and Cell Ranger for RNA mapping, and cells were then demultiplexed as previously reported[19,24,53] (see Supplementary Note for additional details on demultiplexing, see supplementary Data 1 for a list of antibodies used in this study).

**Healthy donor CITE-seq data analysis.** Raw CITE-seq data from our prior report[24] were normalized with the dsb package using the default parameters and empty/background droplets as defined by either clear breaks in the protein library size distribution or droplets defined as negative/background by sample demultiplexing with little impact on normalized values (see Supplementary Note and Supplementary Fig. 8). The *denoise.counts* argument was set to *TRUE* which carries out the recommended step 2 (denoising cell–cell technical variations by estimating and regressing out the technical component for each cell) and the *use.isotype.control* argument set to *TRUE* (defining each cell's technical component by combining isotype control antibody values and the mean of background counts as detailed above). See section below "Assessment of performance of dsb vs. CLR" for methods related to normalization comparisons. Uniform manifold approximation projection[54] (UMAP) was run with the umap-learn Python package in R using reticulate with parameters *n.neighbors = 35, min.dist = 0.6*. Unsupervised protein-based clustering was performed using Seurat[55] to implement the SLM[56] algorithm as we previously reported[24] directly on a distance matrix formed on the protein vs. cells matrix of CITE-seq proteins (without isotype controls) after normalizing with

dsb (in our original report, using pseudocount 1). We retained these cell type annotations used in the original report but renormalized data for all analysis in this paper using dsb with the current package default pseudocount = 10 which resulted in identically distributed relative protein values across cell clusters (Supplementary Fig. 6c, see also Fig. 5c in Kotliarov et al. 2020).

**Assessment of performance of dsb vs. CLR.** Our CITE-seq PBMC data of ~53,000 cells from healthy donors profiled with 83 phenotyping proteins and 4 isotype controls (as shown in Figs. 1 and 3, from Kotliarov et al.) was used for comparison of CLR and dsb normalization using statistical tests, cell type annotation from protein based clustering and comparison of multimodal mRNA + protein-based clustering. For comparisons, the default implementation of dsb, with *denoise.counts = TRUE* and *use.isotype.control = TRUE*, was compared to the CLR transformation across cells, parameters *normalization.method = CLR* and *margin = 2* in the *NormalizeData()* function in Seurat version 4[31]. The Gap statistic[29] for dsb and CLR normalized data was calculated based on $k$ medoids clustering algorithm with $k$ values from 1 to 20, using with 20 bootstrap samples to obtain the reference null distributions. Differential expression testing of protein markers comparing each cluster to all other clusters was performed for the major cell types in the coarse clustering (clusters C0–C10) as reported in Kotliarov et al.[24] vs. all other cells using the *FindMarkers()* function in Seurat to implement a Wilcox test with a log-fold change threshold of 0.3. See section below "Weighted nearest neighbor analysis of CITE-seq and TEA-seq data" for information on clustering comparison.

**Assessment of dsb on external CITE-seq (protein + mRNA) datasets.** Raw and filtered UMI matrices for RNA and ADT counts from Cell Ranger were downloaded from the 10X Genomics website. Background droplets and cells were defined and the default dsb normalization was carried on each dataset as described in the tutorial in the "dsb package documentation [https://github.com/niaid/dsb]". Cells were defined as barcodes in the *filtered* Cell Ranger output, and background drops were defined as after removing the cells from the from the *raw* Cell Ranger output, where a range of ~5e4–7e4 background droplets containing protein reads were used to measure ambient background. Background drops could be clearly differentiated from cell containing droplets by an order of magnitude difference in the protein library size distribution (see blue vs. orange distributions in Supplementary Fig. 4a, h, o). The droplets from these populations were then subjected to standard scRNAseq quality control metrics based on mRNA content, mitochondrial read proportion and protein library size with filters tuned to each dataset in order to retain only high-quality cells in the cell protein matrix and to remove potential cells from the background protein matrix. The number of cell-containing droplets after QC was consistent with the expected per-lane cell recovery based on the cell loading density of the experiment. Proteins with very low raw data signal (a maximum UMI count <5 across all cells) were removed prior to normalization, resulting in removal of the CD34 protein from two datasets. After these basic quality control steps, dsb normalization was carried out using default parameters in the dsb package (*denoise.counts = TRUE and use.isotype.control = TRUE*). Cells were clustered on a cell by protein Euclidean distance matrix of dsb normalized values not including isotype control proteins as described above. UMAP was run with *n_neighbors* parameter = 40 and *min_dist* parameter = 0.4. Cluster labels reflect graph-based clustering in Seurat with resolution tuned to each dataset.

**Assessment of dsb on external proteogenomic (protein + DNA mutation assay) data.** The Mission Bio example data was downloaded from the company's website. Since this dataset only analyzed 10 surface proteins, we performed ambient noise removal, rescaling based on counts in the observed empty droplets only (i.e. performing step 1 only by setting the denoise.counts argument to *FALSE*). UMAP was run with the *min_dist* parameter set to 0.4 and the *n_neighbors* argument set to 40 directly on dsb normalized protein values. Clustering was done on a Euclidean distance matrix using Seurat with a resolution parameter set to 0.5 as described above.

**Analysis of ASAP-seq (protein + chromatin accessibility) and TEA-seq (protein + mRNA + chromatin accessibility) data.** ASAP-seq and TEA-seq data were downloaded from GEO and preprocessed according to the workflows provided in the publicly available analysis code from the original manuscripts. Cell containing droplets were defined as the droplets that passed the authors original quality control metrics. For dsb normalization, we subset non-cells from the raw protein data, estimating noise from the major peak in library size distribution, with quality control to eliminate potential cells from the background matrix, following a similar procedure outlined in the dsb documentation with some modification for ASAP-seq data where mRNA data are not available thus background was estimated based on protein alone from the subset of droplets that did not pass the authors quality control metrics for cells. For comparison, in both datasets cells were normalized with CLR (across cells, margin = 2 in the *NormalizeData()* function using Seurat) and for TEA-seq, an additional log transformation with library size scaling factors (*NormalizeData()* function with parameter *normalization.method = "LogNormalize"*). Analysis of differentially expressed genes in specific clusters vs. all

other cells was carried out in Seurat with the function *FindMarkers*() using an ROC test. Gene Set enrichment analysis of the MAIT cell signature was performed with the fgsea package[57] based on genes ranked by the log2-fold change of genes in cluster 14.

**Weighted nearest neighbor analysis of CITE-seq and TEA-seq data**. For multimodal clustering of the TEA-seq and CITE-seq healthy donor datasets, we used the weighted nearest-neighbor algorithm[31] with the Seurat function *Find-MultimodalNeighbors*(), with slight modification. In pilot analysis of the WNN algorithm we found both CLR and dsb joint embeddings and clustering improved by using protein data directly instead of compressing protein data into principal components. We used the normalized values of 45 (TEA-seq) and 69 (our healthy donor CITE-seq data) phenotyping proteins directly in the joint model (we first removed 14 uninformative/poor performing proteins that had very low average dsb values across all protein-based clusters from the protein data matrix from the healthy donor CITE-seq data). We compared the same joint clustering approach with the only difference being the normalization used in the input data. In analysis of both the CITE-seq and TEA-seq datasets, the mRNA data was compressed into 30 principal components; the same 30 mRNA principal components were combined with either dsb or CLR normalized protein data for joint clustering. First, mRNA data were normalized with the Seurat function *NormalizeData*() with the parameter *normalization.method = "LogNormalize"*, implementing a natural log transformation, standardizing by the library size and multiplying values by 1e4. These values were compressed into 30 principal components based on scaled values for variable genes selected by setting the *FindVariableFeatures*() function with the *selection.method* parameter set to 'vst'. For the CLR WNN model, protein data were normalized by the CLR across cells (using the Seurat function *NormalizeData*() with normalization.method = "CLR" and margin = 2). For the dsb WNN model, data were normalized using the default implementation of dsb, (parameters *denoise.counts = TRUE, use.isotype.control = TRUE*). Seurat was then used to separately cluster the two weighted nearest-neighbor graphs constructed from mRNA principal components and either CLR or dsb normalized input protein data.

**Reporting summary**. Further information on research design is available in the Nature Research Reporting Summary linked to this article.

## Data availability

Raw data used in this analysis are available to download at datasets [https://doi.org/10.35092/yhjc.13370915]. The public datasets included in the data repository were downloaded online and are also available from 10X genomics [https://support.10xgenomics.com/single-cell-gene-expression/datasets] and from Mission Bio [https://missionbio.com/capabilities/dna-protein/#Data]. ASAP-seq and TEA-seq datasets were downloaded from GSE156477 and GSE158013, respectively. All other relevant data supporting the key findings of this study are available within the article and its Supplementary Information files or from the corresponding author upon reasonable request.

## Code availability

The dsb software package is available for download on CRAN: [https://cran.r-project.org/package=dsb]. An analysis workflow with R code and detailed instructions to reproduce the analysis results reported in this manuscript are available for download from github [https://github.com/niaid/dsb_manuscript/][58].

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

## Acknowledgements
This work was funded by the intramural research program of NIAID, NIH (1ZIAAI001152). The authors thank members of the Tsang lab and CHI for feedback on this work, early users who provided valuable feedback on GitHub, Can Liu for testing the dsb package, Yuri Kotliarov for helpful discussions and data deposition assistance related to this work, Sarah Hopkins for assistance with illustrations, and Lucas Graybuck for assistance with TEA-seq data pre-processing.

## Author contributions
M.P.M., A.J.M. and J.S.T. designed experiments, devised analysis strategies, interpreted data, and developed the dsb method. A.J.M. and M.P.M. performed CITE-seq experiments. M.P.M. created the dsb R package and performed analysis. All authors contributed to the writing of the manuscript.

## Funding

## Competing interests
The authors declare no competing interests.
