## [Peer Review File · Nature Communications]

Normalizing and denoising protein expression data from droplet-based single cell profilingReviewers' Comments:

Reviewer #1:

Remarks to the Author:

In Mulè, Martins, and Tsang, the authors introduce *dsb*, a novel R package for the computational method for the denoising and preprocessing of CITE-seq count data. The key statistical advance utilizing two different sources of variation inherent to the experiment are clear and coherent, and the authors do an impressive job at describing and benchmarking how these sources impact CITE-seq data. I would go as far to say that this is one of the best computational papers analyzing/describing CITE-seq that I've seen, and I learned a tremendous amount about these data from the authors. Further, the supplemental and online resources are superb, and I appreciate the level of detail that went into making this tool usable and available to the community. I commend the authors for this important and technically robust work.

Overall, I have no major comments regarding the technical validity of the work. However, as CITE-seq is a fairly mature technology (at least by single-cell genomics standards), existing methods (i.e. the Seurat CLR normalization) have been entrenched in many workflows. Figures 1h/2r are extremely impressive in helping establish CD4 as an epitope with 'intermediate' expression, which I think starts to establish why one would want to utilize *dsb* over other the current options. I would recommend the authors consider the following options to best showcase the utility of the tool.

- Recent methods have extended CITE-seq to include scATAC-seq and/or Multiome capture with oligo tagged antibodies (ASAP-seq / ICICLE-seq / TEA-seq). There is a rationale then to reframe the work from application of CITE-seq to any "droplet-based single-cell proteogenomic" data input, though CITE-seq is a far more parsimonious name. I'd encourage the authors to a) consider reframing the utility of *dsb* to be inclusive of these (and probably other emerging) technologies and b) show a successful application of *dsb* to these datasets.

- Related to the point above, it may be particularly useful for the authors to demonstrate an analysis of ICICLE/ASAP-seq data where the number of tags is decreased due to the treatment conditions of the cell. Further, it appears that the extra washing steps in ASAP-seq decreases some of the background signal (an anecdotal experience), which would be interesting to characterize using the statistical models in *dsb*.

- Recently described antibody panels (e.g. Hao et al. / Mimitou et al. bioRxiv 2020; Triana et al bioRxiv 2021) scale CITE-seq (or related) panels up over 200 antibodies. A demonstration on one or more of these datasets would be very useful for the readers. In particular, many of the antibody clones seem to not work at all in these panels. An application of *dsb* to describe a) poor performing clones but ultimately b) new insights afforded by *dsb* + large antibody panels would be extremely compelling.

- There is currently limited quantitative benchmarking of *dsb* to other methods. In particular, totalVI has emerged as a very promising method for CITE-seq data. Though I appreciate (also in the discussion) that *dsb* and totalVI do different things, a more formal head-to-head comparison would be extremely useful for the readers/ users of the technology.

- Along these lines, could the authors provide a more concrete/quantitative head to head comparison of *dsb* against CLR and related methods? One potential strategy would be to identify clusters of cells using the RNA data and then performing ROC analyses for specific epitopes that should distinguish all cells 'positive' for those markers (e.g. all clusters that are clearly CD4 T cells).

- Ultimately (and I appreciate that this is challenging), there wasn't an extremely strong 'new' biological finding from the reanalysis of the data presented to clearly demonstrate why *dsb* is a superior option from current workflows. Any stronger biological conclusions from the existing or newly

suggested datasets would likely cement dsb as a widely-used tool.

Finally, I acknowledge that there is already a substantial amount of data analysis present in the paper and do not want more analysis for the sake of doing it—my hope though is that one or more of these applications does indeed extend the reach of the tool. Many of the above points may be satisfied with a more detailed discussion in the text as the authors deem appropriate.

Caleb Lareau

Reviewer #2:

Remarks to the Author:

In the paper, authors developed computational methods for normalizing and denoising protein expression data from droplet-based single cell profiling. When CITE-seq users encounter the noise issues, this method of denoising and normalization can improve signals to background noise. This method can be very useful for other single cell analysis studies. I have suggestions/questions which may improve this manuscript.

1. dsb model assumes proteins to be expressed in bimodal distribution. However, many samples including solid tumors may exist highly heterogeneous expression surface proteins. In this case, applying a 2-component Gaussian mixture model seems to be inappropriate. How can your model be applied in these heterogeneous samples? Have authors applied the dsb model for denoising and normalization of CITE-seq data from tissue samples?

2. This paper lacks the assessment of accuracy of a normalization model compared to the others. Can authors devise a metric for comparing the cell type classification accuracy of dsb with joint probabilistic models of mRNA and protein? I am wondering whether dsb outperforms other protein normalization methods.

Reviewer #3:

Remarks to the Author:

Mule et al. describe two sources of protein expression noise in CITE-seq data occurring at two levels: 1) protein-specific noise arising from ambient, unbound antibodies and 2) droplet/cell-specific noise. Normalization of such unwanted sources of noise is a critical step in CITE-seq data analysis, and the authors propose a method to denoise and normalize such data. We have the following comments.

1. The proposed normalization method contains two components. The first part relies on the use of multiple isotype controls. While it is good practice to have such controls in CITE-seq experimental design, they are not always available. The proposed method is further complicated by the reliance on multiple isotype controls since each of them is found to be quite noisy (Fig. 1e). Therefore, the applicability of the first normalization component is highly experiment/data dependent and may not be reliable/applicable when the number of isotype controls is limited (say, only 1) or not available.

2. The second component of the normalization method relies on a 2-component mixture modeling procedure accounting for protein-specific noise. This mixture modeling is applied to each cell across the surface proteins (ADT) included in the libraries for finding these arise from true signal and those that are background noise. The issue with this step is that fitting a robust mixture model requires a large number of ADTs. It is not clear how robust the results are when this is applied to datasets with a small ADT library, and this could be a limitation to the utility of this normalization component.

3. The authors mentioned that current methods are not appropriate for ADT data normalization

because “1) current methods/experiments still measure only a small fraction of unique proteins with a wide range of antigen density on target cell types, resulting in individual protein counts in single cells of less than 10 to more than 1000”. However, the proposed method does not seem to deal with this issue either. How would the proposed method be more appropriate than common normalization procedures in this aspect?

4. From the example CITE-seq datasets, it is not clear what downstream analyses are impacted by the two protein noise sources and how much impact they have on each of these downstream analyses. Could the author numerically quantify these and show how much and how significant the improvements are from using the two proposed normalization steps?

5. The authors mentioned, “... those for protein count data are in their infancy despite the need for such approaches given the substantial levels of noise reported in raw protein counts [ref. 2]”. We wonder how would reference 2, which is about power study of limma on microarray data, support this argument?

Response to reviewers

We thank the reviewers for their appreciation for our work and constructive feedback, which have helped improve our manuscript. Below we respond first to common questions followed by a point-by-point response to address the reviewers' comments.

Common responses

R0.1 – Comparison with the centered log ratio (CLR) normalization

Following the reviewers' suggestions on comparing our method with existing approaches for denoising and normalizing CITE-seq data, we added benchmarking against the centered log ratio (CLR) normalization, which is, as reviewer 1 noted, currently the most used method and the default CITE-seq protein normalization implemented in the popular single cell analysis software package Seurat. Note that the implementation of the CLR has been updated in the most recent version of Seurat: the new version uses across-cell normalization, while an across-protein approach was used previously. We have therefore compared dsb with this new implementation of the CLR normalization.

To assess dsb vs. CLR normalization, we use three orthogonal quality metrics that captured different desirable qualities for a denoising/normalization method: 1) reduction of technical noise, 2) improving cell clustering quality, and 3) helping to better identify proteins differentially expressed between cell clusters. Overall, these analyses consistently indicate that dsb outperformed CLR.

A summary of the findings from each of these assessments is provided below (see also lines 163-182).

1) Technical noise reduction as reflected by reduced variance associated with sequencing depth (see lines 169-173.)

We took advantage of the expectation that within major immune cell subpopulations, variation due to sequencing depth is largely technical as opposed to biological. Using linear models fit within each of the major cell lineages, we found that cells normalized using dsb have reduced variance explained by sequencing depth. This is particularly striking for myeloid cells as shown below. Note that library size/sequencing depth was not a variable used in dsb normalization.

Fig. R0.1.1 (revised manuscript Fig. 2b)

Within each major cell type lineage shown, the percentage of variation in the relative expression level of each surface protein explained by the sequencing depth of each cell (total protein library size); these were estimated from a multivariate model fit to each protein across cells (see methods).

2) Improved cell clustering quality (see lines 173-179.)

Reviewers suggested different ways to assess and compare downstream clustering performance after normalization with dsb; we appreciate these suggestions, however, there are a few challenges in assessing clustering performance improvements with this type of data. First, the use of AUC / accuracy-based approach presents some interpretation challenges. To use clustering accuracy, one must define ‘ground-truth’ cluster labels for each cell. This presents a circular problem for our own data where the cell type labels were defined based on clusters from dsb normalized data. If we use existing cell type annotations from published data, cells that we annotated with potentially *more biologically accurate labels* after processing the data with dsb would appear to be misclassified. For example, as we described in greater detail in R0.2 below, we compared CLR and dsb normalized values as inputs to a new joint clustering model using both mRNA and protein data (the weighted nearest neighbor method: <https://doi.org/10.1016/j.cell.2021.04.048>) generated by a new type of single cell assay called TEA-seq. In this analysis, a population of MAIT cells was revealed only by dsb normalization. This highlights a key limitation in using cell type classification accuracy as a metric for normalization performance. Had we used as “ground truth” the original authors’ cell annotations, which classified these cells in other conventional T cell clusters, these MAIT cells would have been deemed misclassified by the dsb normalized data model, despite their more biologically coherent identity being revealed only after dsb normalization (See **R0.2** and lines 270-307).

Using mRNA data alone for cell type classification accuracy assessment could also be problematic since the concordance between mRNA and protein are known to be poor for many genes, such as in the case of CD4 and CD8 where it is appreciated that mRNA alone cannot robustly separate out CD4+ vs. CD8+ T cells. In general, we are concerned that mRNA based or alternative approaches to define the “ground truth” cell cluster annotation labels would also not bypass the challenges of lacking ground truth on what the expected cell clusters/populations should be, besides annotating the cells based on surface markers via manual gating on normalized values, which would then lead to a circular analysis.

Given the challenges, we decided to bypass the ground truth label approach and instead used a metric designed to assess clustering quality as indicated by the gap statistic (<https://rss.onlinelibrary.wiley.com/doi/10.1111/1467-9868.00293>) to comparatively evaluate dsb vs. CLR normalization. Downstream single cell clustering quality is improved by upstream normalization with dsb as indicated by the gap statistic, which compares within cluster dispersion with that of a reference null distribution as a function of the number of clusters (higher gap statistic indicates better clustering quality). We noted that the trend as a function of k was, however, similar between dsb and CLR, suggesting that the improvement in the gap statistic could be partly due to scaling differences between these two transformations (see also below on the interpretability of dsb scaling.)

Fig. R 0.1.2 (revised manuscript Fig. 2c)

The gap statistic (see methods) for assessment of cluster quality comparing cells normalized using dsb vs CLR (across cells).

3) Improved cell type dependent protein expression estimation (see lines 179-182.)

As discussed in the manuscript, in general the protein expression estimates of dsb are easy to interpret as they reflect the number of standard deviations away from the mean of the expected noise estimated by leveraging information provided by empty droplets, with additional correction for each cell's technical component. Below we further show that the log fold change of dsb normalized protein expression values can better delineate cell type specific expression patterns compared to CLR (Fig. **R0.1.3** below.) This important information for cell cluster/type identification in single cell analysis is thus improved by dsb normalization.

Fig. R0.1.3 (revised manuscript Fig. 2d)

Log fold-change in protein values normalized by dsb compared to CLR (across cells) for a given cell cluster/type compared to all other cell types. Log fold change is increased in dsb normalized values vs CLR for key proteins concordant with known biology of these cell types. Note that the canonical markers in specific cell types (e.g., CD14 on monocytes) tend to have more distinguishable expression from other cells.

RO.2 – broad applicability of and downstream biological insights provided by dsb

An important suggestion from the reviewers was to further demonstrate: 1) the method’s applicability on new types of multimodal single cell assays and 2) the impact of dsb on downstream cell type classification or biological insights obtained.

Addressing the first point, our manuscript now includes additional data on the applicability of dsb to eight external datasets spanning multiple platforms involving single cell protein expression measurements via antibody barcoding (starts at line 184), including data from two new multimodal single cell assays published recently after we submitted our original manuscript (see lines 210-229 and 270-307). We show that the underlying modeling assumptions in dsb are also met in these datasets.

Addressing the second point, in addition to the points addressed in **RO.1** we added analyses (including on the aforementioned new assay types) that demonstrated how dsb can improve downstream analyses and biological interpretations, such as the outcome of joint mRNA and protein clustering (see case study I starting at line 231). For example, misclassified subpopulations can emerge without dsb correction for background noise, whereas using dsb with joint clustering can correctly identify biologically coherent cell subsets/states.

A compelling example involves data from a trimodal single cell assay measuring surface protein via oligo conjugated antibodies, chromatin accessibility (via ATAC-seq), and the transcriptome simultaneously in single cells (TEA-seq data from Swanson et. al. 2021, a method published while our manuscript was under review <https://doi.org/10.7554/eLife.63632>). Compared to CLR, we found that using dsb normalization followed by joint clustering of mRNA and protein (using Seurat) prevented misidentification of a T cell subset, whereas CLR would have led to an incorrect annotation of the cell population (see case study II starting at line 270).

Specifically, dsb normalization followed by joint mRNA-protein clustering revealed a mucosal invariant T cell (MAIT) population that was masked by noise when CLR normalized protein counts were used (see

figure below). In this population, the MAIT T cell receptor (TCR) protein (TCR Valpha 7.2) had high dsb levels (~12), yet their estimated expression on the same cells had near undetectable (<1) CLR normalized values. In the CLR normalized data, the alpha/beta and gamma/delta T cell receptor proteins in this cluster were the highest; we found this reflected CLR's inability to account for noise in empty droplets. Thus, this constitutes a clear example in which dsb enabled recovery of expression of a key marker critical for the identification of a key cell population that would otherwise have been missed with the CLR transformation (see lines 270-307). This example together with our other use case of joint clustering on our own PBMC dataset (see lines 231-268, Figs. 4 f-i) in the revised manuscript illustrate how dsb can enhance biological interpretation of results obtained from joint mRNA-protein clustering using multimodal single cell data.

Fig. R0.2.1 (parts of revised manuscript Fig. 5)

Enhanced analysis of TEA-seq data (transcriptome, chromatin accessibility and surface proteins) using dsb for normalization of protein data. **d**. UMAP plot of single cells and clusters derived by WNN joint mRNA-protein clustering with data normalized using CLR or **e**. dsb. The average protein expression profiles of the clusters are shown below as heatmaps. **f**. Similar to Fig. 4g but here for cluster 14 from (e) using dsb or **g**. CLR. **h**. Differential expression analysis (ROC classifier) of genes in cluster 14 vs other clusters. **i**. Comparison of the log fold change values for the top MAIT-specific differentially expressed genes from FACS-sorted TCR-va7.2+/MAIT cells compared to other T cells (RNA-seq data from Park *et. al.* 2019) with the log fold change of these genes in the TEA-seq dataset in cluster 14 cells vs other cells.

R0.3 – Clarification of the dsb algorithm and denoising steps

Some of the reviewer's concerns appear to arise from a misunderstanding of the dsb algorithm steps, which could have been presented more clearly in our manuscript; we have now clarified the algorithm in the text (e.g., see lines 59-65, 138-144) and methods (lines 382-445); we also provide a summary below. This addresses points raised about the single cell Gaussian mixture models and the inclusion of isotype controls used in the optional (but recommended) second step of dsb.

As a recap, dsb consists of two steps: the first is to remove protein-specific noise that we found is driven by ambient antibody capture (which does not involve a mixture model fit, and instead uses information from empty droplets to estimate the background). The second step we are discussing below, with the goal of estimating and removing cell-to-cell technical variations, is optional (but recommended), thus for datasets with either a very small number of proteins or lacking isotype controls, dsb is still applicable since users can use step I only to remove ambient antibody-derived noise.

To further assess the robustness of step II, we have added analyses to show that protein levels within individual cells do not strictly need to have a bimodal distribution in order for the relative magnitude of the background component to be robustly estimated as the first component mean (μ_1) of a 2-component Gaussian mixture model fit (see lines 106-119). For example, cells that fit more optimally with a 3-component Gaussian mixture model have very similar/highly correlated μ_1 parameters whether the protein population is fit with a 2 or 3 component Gaussian model (see new Supplementary Figs. 2b-f; see also Fig. R 0.3.1 below). In addition, we show that the dsb technical component is robust and highly correlated across cells when a 2-component model vs. 3-component model is used for defining μ_1 (Fig. R0.3.1 below); this was true even on datasets which measured 14 proteins only together with 3 isotype controls (see lines 199-209 and Supplementary Fig 2f). This feature is in part due to the robustness of the μ_1 parameter estimate described above, and also because the dsb technical component combines correlated information shared among isotype controls and μ_1 ; this anchors the component of μ_1 that is correlated with noise, improving robustness of the per-cell technical factor estimate compared to if μ_1 was used alone to define the technical component. This is also the main reason we recommend the use of isotype controls in such experiments.

As an additional clarification, the full procedure for the second step can be broken down into *three parts* as described in the methods section (lines 381-444) to further explain this step (see equations II, III, IV, lines 409-440; please see also revised methods section for additional details). As detailed in step II part I, mixture models are only applied within each single cell, not across cells. We extract only the μ_1 parameter (the mean of the lowest expressing population of proteins within a cell/droplet) from each single cell Gaussian mixture model, then *combine* this value with signals from multiple isotype controls (in step II part II) to derive the technical factor (i.e., the correlated signal between μ_1 and the isotypes). Please see also the Supplement text section "Importance of using isotype control antibodies for estimating cell-intrinsic normalization factors" and Supplementary Figs. 2 and 3).

Reviewers also commented that the 2-component Gaussian (step II part I above) might not be appropriate when profiling heterogeneous tissues like tumor samples. As explained above, the second step is applied within each cell; the main requirements are therefore that the staining panel is sufficiently diverse to contain a group of protein markers negative in each cell (for μ_1 estimation), and ideally also includes non-targeting isotype controls; such panels are more likely to be useful in heterogeneous samples. A concern with a per-cell background correction in a mixture with diverse cell types/states is the inadvertent removal of biological differences between cells; as mentioned in the main text (lines 87-238) and elaborated upon in the Supplementary Note (lines 122-139), dsb's approach of only using the shared information between μ_1 and the isotype controls as the estimate of

the per-cell technical factor for removal was designed to mitigate the risk of removing biological signals because biologically relevant signals could still be contained within the μ_1 component alone, even in diverse cell mixtures. Another caveat highlighted in our supplementary note is that cell types more prone to higher non-specific antibody binding, such as monocytes, may receive “over” correction during this step. We recommend that this issue needs to be considered in the experimental protocol, for example, by ensuring appropriate Fc receptor blocking prior to staining.

Fig. R 0.3.1 (revised manuscript Fig. 1i)

Pearson correlation between the dsb technical component as calculated using the μ_1 parameter extracted from a 2-component Gaussian mixture model (x-axis) vs a 3-component mixture model (y-axis).

The data used in the figure above was our PBMC dataset measuring 87 proteins. To address another point raised by a reviewer that the 2-component mixture may not be appropriate on smaller panels, in our revised manuscript we provide analysis of an independent CITE-seq dataset measuring only 14 proteins and 3 isotype controls. We found that even with this small number of proteins and independent of whether a 2-component or 3-component Gaussian mixture was used to estimate μ_1 , the value for the resulting technical component was highly correlated (lines 199-209 and see figure below). In addition, the technical component as calculated (in the default implementation of dsb) using a 2-component mixture was also correlated with the protein library size *within* each cell cluster, which is to be expected as differences in the library size across cells are known to capture aspects of technical noise. This additional analysis further supports that the default dsb algorithm can be applied as intended in datasets with even small protein panels. Since we have not evaluated a dataset with even smaller protein panels, we added a note in our software documentation that users with very small panels can carry out similar robustness checks on their data before running the second step of dsb. To facilitate this, we added functionality in the dsb R package to return the relevant statistics for such assessments.

Fig. R 0.3.2 (revised manuscript Supplementary Fig. 2f and Supplementary Fig. 4d)

Left panel: as shown above in R 0.3.1, for the PBMC 10K V3 dataset measuring 14 phenotyping proteins and 3 isotype controls. **Right panel:** the dsb technical component Pearson correlation with the protein library size within cell clusters defined by surface protein.

Finally, we also gained additional insights from reviewer 1’s suggestion to test the dsb model assumptions on ASAP-seq data (single cell ATAC-seq plus proteins measured with barcoded antibodies as in CITE-seq). Relevant to dsb step II in particular, this dataset profiled more than 200 surface proteins, thus represents an example on the larger end of the spectrum in terms of protein panel sizes. In this dataset we found that the $k=3$ component model had a better fit than the $k=2$ component model in 85% of cells. However, consistent with our observations above, the background mean estimated from the $k=2$ vs. the $k=3$ models were highly concordant and when the estimated μ_1 parameter was combined with the isotype controls, the resulting dsb technical component was again very similar and highly correlated. Thus, even when the $k=3$ fits may be deemed numerically more optimal for many cells, the background mean and the technical component can still be robustly extracted via a $k=2$ component model. We also assessed the correlation between library size and the technical component inferred by the $k=2$ vs. the $k=3$ fits. They were correlated in both cases, with the $k=3$ fit generally resulted in a lower correlation than the $k=2$ component model (Pearson correlation 0.51 vs 0.62). These data together support that the default dsb implementation can be kept unchanged for protein panels of different sizes.

Note that dsb was also applied in our recent work on COVID-19 involving a panel containing over 190 antibodies: <https://www.ncbi.nlm.nih.gov/pmc/articles/PMC7874909/>

In addition, dsb was used in other recent papers using large CITE-seq antibody cocktails, e.g., <https://www.nature.com/articles/s41586-021-03929-x> <https://www.medrxiv.org/content/10.1101/2021.05.11.21256877v1.full.pdf>

Note that we were not involved in these works and have no relationship to the authors.

Reviewer #1 (Expertise: The analysis of CITE-Seq and other single cell data):

In Mulè, Martins, and Tsang, the authors introduce dsb, a novel R package for the computational method for the denoising and preprocessing of CITE-seq count data. The key statistical advance utilizing two different sources of variation inherent to the experiment are clear and coherent, and the authors do an impressive job at describing and benchmarking how these sources impact CITE-seq data. I would go as far to say that this is one of the best computational papers analyzing/describing CITE-seq that I've seen, and I learned a tremendous amount about these data from the authors. Further, the supplemental and online resources are superb, and I appreciate the level of detail that went into making this tool usable and available to the community. I commend the authors for this important and technically robust work.

Overall, I have no major comments regarding the technical validity of the work. However, as CITE-seq is a fairly mature technology (at least by single-cell genomics standards), existing methods (i.e. the Seurat CLR normalization) have been entrenched in many workflows. Figures 1h/2r are extremely impressive in helping establish CD4 as an epitope with 'intermediate' expression, which I think starts to establish why one would want to utilize dsb over other the current options. I would recommend the authors consider the following options to best showcase the utility of the tool.

R1.0 We thank this reviewer for their careful reading of our manuscript and appreciation for our work.

We have also recently significantly updated the documentation for the tool to simplify and improve the usability of our tool for more novice users <https://github.com/niaid/dsb>.

Indeed, by far the most common normalization method used on CITE-seq data in current publications is the Seurat default centered log ratio (CLR) transformation. Based in part on this suggestion, we chose to compare dsb to the latest version of the CLR transformation throughout the benchmarking analysis added to the manuscript (see also **R0.1** above).

- Recent methods have extended CITE-seq to include scATAC-seq and/or Multiome capture with oligo tagged antibodies (ASAP-seq / ICICLE-seq / TEA-seq). There is a rationale then to reframe the work from application of CITE-seq to any "droplet-based single-cell proteogenomic" data input, though CITE-seq is a far more parsimonious name. I'd encourage the authors to a) consider reframing the utility of dsb to be inclusive of these (and probably other emerging) technologies and b) show a successful application of dsb to these datasets.

R1.1 Thank you for this suggestion; indeed, for simplicity, we used the term CITE-seq to refer generally to "feature barcoding" data, a term used by the company 10X genomics (i.e., data shown in Fig. 2, Supplementary Figs. 5,6), and "Proteogenomic profiling" data, a term used by Mission Bio for their platform for DNA and surface protein assessment (shown in Supplementary Fig. 7 in the original submission). To avoid confusion, in our revised manuscript, we used the term "CITE-seq", coined in the original NY Genome Center publication, to refer to transcriptome + protein profiling data including "feature barcoding" data from 10X genomics, and we refer to the new assay technologies by the names proposed in their respective manuscripts (TEA-seq and ASAP-seq). We feel that our title referring to "protein expression data from droplet-based single cell profiling" is inclusive of all these data types. To further clarify, we also added text and additional analysis demonstrating that dsb is compatible with these data types and will likely continue to be applicable to future developments in single cell assays

using barcoded antibodies for protein expression profiling, since the noise sources we revealed in this manuscript seem to be shared across these different droplet-based assays (see **R0.2** above).

- Related to the point above, it may be particularly useful for the authors to demonstrate an analysis of ICICLE/ASAP-seq data where the number of tags is decreased due to the treatment conditions of the cell. Further, it appears that the extra washing steps in ASAP-seq decreases some of the background signal (an anecdotal experience), which would be interesting to characterize using the statistical models in dsb.

R1.2 We analyzed ASAP-seq data (<https://doi.org/10.1038/s41587-021-00927-2>) and indeed found lower background noise in this dataset; however, dsb normalization remained compatible as our modeling assumptions were met (Supplementary Figs. 5 d-j); see also lines 212-229. Using dsb for normalization on this dataset remained helpful in identifying true staining from noise. Using the same threshold (3.5 – corresponding to 3.5 standard deviations from the background droplet mean) we used throughout all datasets in the manuscript, we saw clear delineation of positive and negative populations in this bone marrow dataset. With CLR such a threshold would have to be set for each protein separately to adjust for each protein’s unique noise floor.

Fig. R1.1 (revised manuscript Supplementary Figs. 5d-j)

Analysis of ASAP-seq data (protein assessment as in CITE-seq with chromatin accessibility assayed instead of the transcriptome). d. background droplets and cells defined by protein and mRNA library size. e. As in Figure 1g, correlation matrix of variables comprising the dsb technical component. f. As in Fig 1h, isotype control mean vs background mean per cell. g. As in Figure 2g, relationship between protein library size and the dsb technical component. h. UMAP projection and clusters based on dsb normalized protein values. i. Biaxial plot of CD3 vs CD4 with the dsb threshold of 3.5 applied. j. As in (f), with data normalized with the CLR transformation (across cells).

- Recently described antibody panels (e.g., Hao et al. / Mimitou et al. bioRxiv 2020; Triana et al bioRxiv 2021) scale CITE-seq (or related) panels up over 200 antibodies. A demonstration on one or more of these datasets would be very useful for the readers. In particular, many of the antibody clones seem to not work at all in these panels. An application of dsb to describe a) poor performing clones but ultimately b) new insights afforded by dsb + large antibody panels would be extremely compelling.

R1.3 As detailed also in **R0.3** above, in addition to the analysis detailed in **R1.2**, we further looked into the intermediate modeling steps used by dsb, which helped our understanding regarding the performance and applicability of dsb on datasets measuring many (i.e., over one hundred) proteins. The ASAP-seq analysis in the revised manuscript provides an example of dsb's compatibility with a large antibody panel targeting more than 200 proteins. We also added text referencing our lab's recently published work on Covid 19 where we stained cells with an antibody panel of 192 proteins and used dsb for normalization: <https://www.ncbi.nlm.nih.gov/pmc/articles/PMC7874909/>.

With respect to poor performing clones, we and others have also observed this. Clones with low or no read counts (e.g., a maximum UMI count across all cells of ~5 or less) likely cannot be successfully recovered by normalization methods. Such data can originate from clones that stain a target with extremely low surface antigen density, weak antibody binding, incorrect staining concentration, or the lack of the targeted cell population in the experiment. We recommend users remove protein with no evidence of staining in the raw data – we have updated our documentation to reflect this suggestion, please see <https://github.com/niad/dsb#step3> “Optional step; remove proteins without staining”. Recent work from Buus *et. al.* (<https://elifesciences.org/articles/61973>) also suggested if a protein target does not respond to antibody dilution, positive signal may be absent from the cells being stained and thus one should reevaluate whether to include such an antibody in the experiment. If such low staining targets are left in the analysis, the reviewer's intuition is correct that after dsb normalization, these non-staining clones have values that indicate they are not significantly different from background (i.e., less than the threshold of 3.5 we used). In practice, this has allowed us to exclude these from further consideration in our own previous work where we did not remove them prior to dsb normalization.

Another possibility presented in Buus *et. al.* was that these low staining targets could still point to expressed proteins but obscured by background noise. The example above using TEA-seq data (common response to reviewers **R0.2**) involving the MAIT cell population is such an example, where dsb helped recover a low-staining antigen TCR Valpha 7.2. This signal was notably absent from any population after CLR normalization but appeared specifically in the MAIT population after dsb normalization. The authors of the TEA-seq manuscript did in fact considered this TCR a poorly performing antibody, though with dsb we recovered the biologically relevant signal.

- There is currently limited quantitative benchmarking of dsb to other methods. In particular, totalVI has emerged as a very promising method for CITE-seq data. Though I appreciate (also in the discussion) that dsb and totalVI do different things, a more formal head-to-head comparison would be extremely useful for the readers/ users of the technology.

R1.4 Please see common response **R0.1**. Please note that we focused on comparing dsb to the centered log ratio (CLR) transformation because as the reviewer had noted, CLR is by far the most used method for normalizing CITE-seq data.

Regarding probabilistic models like TotalVI, we further clarified (lines 318-334) how dsb is designed to be compatible with higher level methods e.g., we designed dsb to be directly compatible with any downstream analysis task including dimensionality reduction clustering, differential expression, and other types of statistical modeling. Our intended users are thus those using statistical / machine learning algorithms provided by tools within comprehensive toolkits, such as those provided by Seurat, Bioconductor, or Scanpy. For example, we demonstrated how unsupervised statistical approaches such as Seurat's weighted nearest neighbor joint clustering method are improved by using dsb first for normalization (see Figs. 4f-i and 5e-l).

In contrast, the goal of TotalVI is to provide an end-to-end Bayesian probabilistic deep learning latent space model, rather than to provide transformed data compatible with other downstream analysis task. Importantly, TotalVI does not normalize protein data, but returns posterior probability distributions for protein expression (i.e., with values bounded between 0 and 1) and 'denoised' raw values designed for Bayesian count modeling. In order to be used outside of TotalVI's intended end-to-end framework (such as comparing to our method), these counts would require further processing and normalization. Thus, these counts and the posterior probabilities returned by TotalVI are not directly compatible with the type of head-to-head benchmarking analysis we have performed (e.g., the gap statistic, differential expression, and mixed linear models – see **R0.1** above) for assessing dsb against the CLR transformation.

Probabilistic models such as TotalVI rely on modeling the data / noise generation process and could potentially be enhanced by incorporating direct measures of noise available in the data, as performed by dsb based on the experiments and analysis we performed to investigate the noise generation process. Indeed, direct incorporation of the ambient component of noise that we discovered and described in our preprint as a key driver of protein-specific noise has been suggested as a potential improvement to TotalVI (<https://github.com/YosefLab/scvi-tools/issues/684> -- we have no relationship to the researcher making this suggestion). Thus, we believe that while comparing between analysis strategies like neural network vs. our approach that more directly uncover and estimate both protein- and cell-specific noise from CITE-seq data could be an interesting topics for further study, it is outside the scope of our already lengthy experimental and computational analyses involving joint experimental-computational examination of noise sources, assessment of dsb's model assumptions and utility in eight independent datasets, and benchmarking against the most commonly used and directly comparable method (CLR).

- Along these lines, could the authors provide a more concrete/quantitative head to head comparison of dsb against CLR and related methods? One potential strategy would be to identify clusters of cells using the RNA data and then performing ROC analyses for specific epitopes that should distinguish all cells 'positive' for those markers (e.g. all clusters that are clearly CD4 T cells).

R1.5 Please see common response **R0.1**. Thanks to the reviewer's suggestion we systematically compared dsb to CLR. For reasons discussed in **R0.1**, we opted for an unbiased approach to define clustering quality (via the gap statistic). We also highlight examples where joint protein-mRNA clustering with the weighted nearest neighbor algorithm is enhanced by dsb compared to CLR normalization (See **R0.2**, and Figs. 4f-i and Fig. 5).

- Ultimately (and I appreciate that this is challenging), there wasn't an extremely strong 'new' biological finding from the reanalysis of the data presented to clearly demonstrate why dsb is a superior option from current workflows. Any stronger biological conclusions from the existing or newly suggested datasets would likely cement dsb as a widely-used tool.

R 1.6 Please see common response **R0.2**. In addition to the TEA-seq example where we revealed a MAIT cell population not found using CLR normalization, we included an example of a population we interpreted as an inflammatory / activated monocyte based on weighted nearest neighbor analysis of our PBMC data. We believe these examples provided in this revision illustrate the benefits of dsb in enhancing the identification and annotation of key cell populations, which are important for downstream biological interpretations.

Finally, I acknowledge that there is already a substantial amount of data analysis present in the paper and do not want more analysis for the sake of doing it—my hope though is that one or more of these applications does indeed extend the reach of the tool. Many of the above points may be satisfied with a more detailed discussion in the text as the authors deem appropriate.

We thank the reviewer again for their thoughtful comments that helped to improve our manuscript significantly.

Reviewer #2 (Expertise: Disruptive single cell technology development):

In the paper, authors developed computational methods for normalizing and denoising protein expression data from droplet-based single cell profiling. When CITE-seq users encounter the noise issues, this method of denoising and normalization can improve signals to background noise. This method can be very useful for other single cell analysis studies. I have suggestions/questions which may improve this manuscript.

1. dsb model assumes proteins to be expressed in bimodal distribution. However, many samples including solid tumors may exist highly heterogeneous expression surface proteins. In this case, applying a 2-component Gaussian mixture model seems to be inappropriate. How can your model be applied in these heterogeneous samples? Have authors applied the dsb model for denoising and normalization of CITE-seq data from tissue samples?

R2.1 Regarding the 2-component mixture fit within individual cells, please see **R0.3** in the common response above that clarified step II of our algorithm. Regrettably, the way we originally described step II gave the impression that it was strictly required for cells to have a bimodal protein count distribution. In our original manuscript, we have assessed the robustness of this step by examining the goodness of fit of the μ_1 parameter extracted from the Gaussian mixture with different number of mixture components (we tested 1-6). We found that in cells with for example, a 3-component model as the best fit (as evaluated by the BIC), the μ_1 parameter was highly correlated with the 2-component fit. In our revised manuscript, we performed additional analysis to show that the resulting dsb technical component across cells (the shared variation in μ_1 and isotype controls) is also highly concordant with that extracted when the μ_1 parameter was estimated from either 2 or 3 component fits (further detailed in

common response **R0.3**). Based in part on these analyses we reframed the text to explicitly state that a bimodal protein distribution within single cells is not required.

We have not performed tissue CITE-seq experiments ourselves. In addition to the clarifications provided in **R0.3**, we refer the reviewer to the following manuscript in which CITE-seq was applied to cells from intestinal mucosa where the authors used dsb to normalize their data <https://www.biorxiv.org/content/10.1101/2021.03.28.437379v1.full.pdf>. Note that we do not know the authors and were not involved in this work.

2. This paper lacks the assessment of accuracy of a normalization model compared to the others. Can authors devise a metric for comparing the cell type classification accuracy of dsb with joint probabilistic models of mRNA and protein? I am wondering whether dsb outperforms other protein normalization methods.

R2.2 Please see common response to reviewers above (**R0.1** and **R0.2**), where we addressed this comment by comparing dsb normalization with CLR using three orthogonal metrics. The downstream performance of dsb normalization was also assessed using data from a new trimodal assay via an joint mRNA + protein clustering approach from Seurat, which provided an example where a key cell population was revealed correctly only with dsb normalized protein values as input to the joint clustering as opposed to when CLR normalized values were used as inputs. We chose to compare dsb normalization to the commonly used method CLR in Seurat rather than probabilistic models that model data at the level of counts for the reasons explained in **R1.4** above. We also chose to focus on unbiased clustering quality metrics for assessment of normalization instead of using cell cluster annotation accuracy due to the reasons outlined on **R0.1**.

Reviewer #3 (Expertise: Analysis of CITE-Seq data):

Mule et al. describe two sources of protein expression noise in CITE-seq data occurring at two levels: 1) protein-specific noise arising from ambient, unbound antibodies and 2) droplet/cell-specific noise. Normalization of such unwanted sources of noise is a critical step in CITE-seq data analysis, and the authors propose a method to denoise and normalize such data. We have the following comments.

1. The proposed normalization method contains two components. The first part relies on the use of multiple isotype controls. While it is good practice to have such controls in CITE-seq experimental design, they are not always available. The proposed method is further complicated by the reliance on multiple isotype controls since each of them is found to be quite noisy (Fig. 1e). Therefore, the applicability of the first normalization component is highly experiment/data dependent and may not be reliable/applicable when the number of isotype controls is limited (say, only 1) or not available.

R3.1 We thank the reviewer for their careful reading of our manuscript and helpful comments.

We wish to first clarify that dsb consists of two steps: 1) defining/removing protein-specific noise and 2) estimating and mitigating cell-to-cell technical variations. Here the reviewer's comments concerned **the second but not the first step of our method**. Please refer also to **R0.3** in the common response above.

As the reviewer pointed out, since isotype controls may not be always available, they are optional and not strictly required for dsb normalization, but rather highly recommended. See also the section in our software documentation on CRAN ("vignettes" tab: <https://cran.r-project.org/package=dsb>) entitled

“Using dsb with data lacking isotype controls”. Users can either run the first step alone (removing protein-specific noise associated with ambient antibody capture, this is our recommendation), or can run both the first and second steps but in the second step only infer each cell’s technical component via the Gaussian mixture model alone. In the TEA-seq dataset (see common response to reviewers **RO.2**), only one isotype control was included. We normalized this data with the default implementation of dsb. In this case, the technical component (derived in step 2) corresponds to the first principal component of two variables across single cells: 1) μ_1 (mean of background protein population); and 2) the isotype control.

In revised Fig. 2a (copied below) we provide an example comparison of data with protein-specific ambient noise removal only (step I – right plot) compared to the application of both steps 1 and 2 (left plot) using isotype controls. As expected, the spiked-in unstained cells (representing the ground truth background population) are centered at 0 in both cases.

Fig. R3.1 (revised manuscript Fig. 2A)

Single cell protein expression of CD4 (x-axis) vs. CD14 (y-axis). Contour lines in red are the distribution of CD4 and CD14 in unstained control cells after normalization in the exact same way as the stained cells in black within each panel, including dsb normalization using the same empty droplets for ambient correction of the unstained cells. The panel on the left corresponds to dsb normalization with both ambient (empty droplet) background correction and removal of technical component λ (step I and II); this is the default implementation of the software (`denoise.counts = TRUE`, `use.isotype.controls = TRUE`). The plot on the right corresponds to ambient (empty droplet) background correction only (step I only, `denoise.counts = FALSE`).

Finally, while dsb is compatible with data without isotype controls, we would like to further stress that having multiple isotype controls is beneficial for robust inference of the technical component for individual cells (dsb step II), as we have now shown in the manuscript (lines 83-146, 206-209 Supplementary Note lines 122-132). We also hope that our work will help promote the inclusion of isotype controls in future CITE-seq experiments, which is also a longstanding best practice in flow cytometry.

2. The second component of the normalization method relies on a 2-component mixture modeling procedure accounting for protein-specific noise. This mixture modeling is applied to each cell across the surface proteins (ADT) included in the libraries for finding these arise from true signal and those that are background noise. The issue with this step is that fitting a robust mixture model requires a large number of ADTs. It is not clear how robust the results are when this is applied to datasets with a small ADT library, and this could be a limitation to the utility of this normalization component.

R3.2 We thank the reviewer for bringing up this important point which we address in the common response to reviewers **R0.3**. We agree the robustness of 2-component Gaussian mixture models will improve with additional observations (in this case proteins in the panel). This concern was precisely why we assessed independent external datasets ranging in protein panel sizes of just 17 proteins to now in the revised version of our manuscript, more than 200 proteins.

This comment refers to what we call step II part I, as described in the common response to reviewers **R0.3**, where we discuss additional robustness checks specifically of the dsb technical component on small (less than 20) vs. large (more than 200) protein panels.

A benefit of CITE-seq is the ability to use larger panels of antibodies than possible in flow cytometry or CyTOF. Multiple recent publications have adopted pre-made panels provided by vendors such as Biolegend that contain more than 100 antibodies (including isotype controls), for example, in our own work on Covid-19 (see lines 224-229) and other recent papers which used dsb for normalization:

<https://www.nature.com/articles/s41586-021-03929-x>

<https://www.medrxiv.org/content/10.1101/2021.05.11.21256877v1.full.pdf>

(We were not involved in these works and have no relationship to the authors.)

While we expect that small panels are used less frequently, we do want to note again that dsb is compatible with even smaller panels (to even down to one antibody) by using step I only, through which the users can still benefit from the removal of protein-specific ambient noise. An example of applying dsb without the second step is shown in Fig. 2a in the revised manuscript and in Fig. R3.1 above.

3. The authors mentioned that current methods are not appropriate for ADT data normalization because “1) current methods/experiments still measure only a small fraction of unique proteins with a wide range of antigen density on target cell types, resulting in individual protein counts in single cells of less than 10 to more than 1000”. However, the proposed method does not seem to deal with this issue either. How would the proposed method be more appropriate than common normalization procedures in this aspect?

R3.3 Our discussion of the normalization challenges associated with protein data obtained from a targeted panel versus transcriptome-wide mRNA data indeed required more clarification. We have expanded the text in the revised manuscript (lines 74-86) and further elaborate below

In the lines quoted in the reviewer’s comment above, we were referring to the commonly applied single cell RNAseq mRNA normalization methods, which involve dividing expression counts by the library size and then multiplying by a global scaling constant. In this case library size is defined as total UMI counts per cell. For example, the Scanpy function `scanpy.pp.normalize_total()`, the Bioconductor (scater package) function `logNormCounts()`, or the Seurat function `LogNormalize()` all apply the same normalization heuristic defining the expression of gene j in cell i is as:

$$\log 1 + \left(\frac{10^5 * UMI_{i,j}}{\sum UMI_i} \right)$$

Here UMI represents the UMI count of gene j in cell i . The correction factor (in the denominator that applied to each cell) captures the total library size.

Given the thousands of transcripts covered in single cell mRNA data the total “library size” tends to better reflect technical factors (e.g., PCR efficiency) affecting the read counts of most transcripts as biologically driven differences typically impact the expression of subsets of genes across single cells. However, in the case of CITE-seq this factor can be highly panel dependent.

For example, examining 6 random cells in CITE-seq data from the original NY Genome Center manuscript (<https://www.nature.com/articles/nmeth.4380>), one can see substantial variations in library size that may depend on the protein panel; this is also consistent with the large dynamic range for protein abundances on cells, compared to sparse mRNA data.

Raw CITE-seq data:

	cell1	cell 2	cell3	cell4	cell5	cell6
CD3	60	52	89	55	63	82
CD4	72	49	112	66	80	78
CD8	76	59	61	56	94	57
CD45RA	575	3943	682	378	644	479
CD56	64	68	87	58	104	44
CD16	161	107	117	82	168	92
CD10	156	95	113	66	129	66
CD11c	77	65	65	44	92	63
CD14	206	129	169	136	164	122
CD19	70	665	79	49	81	44
CD34	179	79	78	83	152	103
CCR5	99	101	85	60	110	50
CCR7	104	72	80	46	89	69
Total Count	1899	5484	1817	1179	1970	1349

The library size as computed above for proteins has contributions from both technical and biological sources – the biological contributions in the raw data above may be substantial (i.e. high CD45RA expression on naïve T cells), so what we were trying convey in the original text quoted by the reviewer was that removing variation due to library size from protein count data would not work well for CITE-seq. In addition, we have shown that such a library size normalization procedure can create distortions and biases in CITE-seq data, e.g., as shown in the figure below where the ground truth noise (red distribution; unstained control cells) is not zero centered after library size correction factors.

Fig. R 3.2 (Revised manuscript Fig. 2A)

Single cell protein expression of CD4 (x-axis) vs. CD14 (y-axis). Contour lines in red are the distribution of CD4 and CD14 in unstained control cells after normalization in the exact same way as the stained cells in black within each panel. With dsb normalization (top panel), the ground truth noise is centered at zero, allowing one to define the two true staining populations with expression above background noise. With a common mRNA normalization procedure applied to CITE-seq data (using library size normalization factors and a global scaling constant), it is unclear which of the populations are truly positive for the two markers, as the ground truth noise cell population has high expression of both markers obscuring the biological signal.

Thus, instead of the library size, we aimed to derive a “more conservative correction factor” to mitigate cell-to-cell technical variations (step 2 of dsb). The way in which this correction factor (the dsb technical component) is more conservative lies in its derivation from isotype controls and non- or lowly staining antibodies in individual cells—the fact that these two independently derived signals are correlated (Figs 1f-g) suggests that the shared variation between them reflect technical rather than biological contributions. Furthermore, this technical factor is being treated as a covariate and regressed out of the protein counts via a linear model rather than as a multiplicative factor applied to protein counts as in library size-based correction (See common response to reviewers **R0.3** and methods).

As expected, given that the library size has both technical and biological contributions, the technical factor derived by dsb is correlated with the protein library size across single cells within cell types/clusters (Supplementary Fig. 3a). The dsb technical factor thus promotes the removal of technical variations while preserving biological differences across cells. We have further clarified these points in our methods section (see lines 409-445).

4. From the example CITE-seq datasets, it is not clear what downstream analyses are impacted by the two protein noise sources and how much impact they have on each of these downstream analyses. Could the author numerically quantify these and show how much and how significant the improvements are from using the two proposed normalization steps?

R3.4 Please see the common response to reviewers **R0.1** and **R0.2**. We thank the reviewer for this suggestion; the manuscript has been improved by adding these comparisons of downstream improvements in clustering and cell type identification, including assessment using joint mRNA–protein clustering from Seurat.

5. The authors mentioned, “... those for protein count data are in their infancy despite the need for such approaches given the substantial levels of noise reported in raw protein counts [ref. 2]”. We wonder how would reference 2, which is about power study of limma on microarray data, support this argument?

R3.5 We thank the reviewer for catching this incorrect citation, which is now corrected in the revised manuscript. The reference was intended to point to a CITE-seq study that reported background noise of unknown origin.

Reviewers' Comments:

Reviewer #1:

Remarks to the Author:

The authors have greatly improved the manuscript in the latest version, showing additional comparisons to the CLR method as well as broader applicability across a range of additional datasets and technologies. I recommend the work as-is for publication.

Reviewer #2:

Remarks to the Author:

The authors have clarified the dsb algorithm step by step both in the main text and methods section. The details of the algorithms in the revised manuscript were better described for the readers to understand than in the original manuscript. The authors evaluated BIC to determine the number of components in the Gaussian mixture model and showed that μ_1 parameters were highly correlated between 2-component mixture model and 3-component mixture model, indicating the technical component is independent of the number of components. Also, as requested, the authors compared dsb normalization method with an existing method, specifically centered log ratio (CLR) normalization and showed that dsb normalization method clearly outperforms CLR by various metrics. This allows researchers to confidently adopt dsb as an alternative normalizing method in droplet-based single cell experiment. I think this revised paper is sufficient to be published in Nature Communications.

Reviewer #3:

Remarks to the Author:

The authors have addressed all my comments. I have no additional suggestions.